# Variations in triple isotope composition of dissolved oxygen and primary production in a subtropical reservoir

Hana Jurikova[1a], Tania Guha[1], Osamu Abe[2], Fuh-Kwo Shiah[1], Chung-Ho Wang[3], and Mao-Chang Liang[1,4,5]

[1]Research Center for Environmental Changes, Academia Sinica, 11529 Taipei, Taiwan
[2]Graduate School of Environmental Studies, Nagoya University, Chikusa, 464-8601 Nagoya, Japan
[3]Institute of Earth Sciences, Academia Sinica, 11529 Taipei, Taiwan
[4]Graduate Institute of Astronomy, National Central University, 32001 Jhongli, Taiwan
[5]Department of Physics, University of Houston, Houston, TX 77004, USA
[a]Now at: GEOMAR Helmholtz-Zentrum für Ozeanforschung Kiel, Wischhofstr. 1-3, 24148 Kiel, Germany

*Correspondence to*: Mao-Chang Liang (mcl@rcec.sinica.edu.tw)

**Abstract.** Lakes and reservoirs play an important role in the carbon cycle, and therefore, monitoring their metabolic rates is essential. The triple oxygen isotope anomaly of dissolved $O_2$ [$^{17}\Delta = \ln(1+\delta^{17}O) - 0.518 \times \ln(1+\delta^{18}O)$] offers a new, in situ, perspective on primary production, yet little is known on $^{17}\Delta$ from freshwater systems. We investigated the $^{17}\Delta$ together with oxygen : argon ratio [$\Delta(O_2/Ar)$] in the subtropical Feitsui Reservoir in Taiwan from June 2014 to July 2015. Here, we present the seasonal variations in $^{17}\Delta$, GP (gross production), NP (net production) and the NP/GP (net to gross ratio) in association with environmental parameters. The $^{17}\Delta$ varied with depth and season, with values ranging between 26 and 205 per meg. The GP rates were observed to be higher ($702 \pm 107$ mg C m$^{-2}$ d$^{-1}$) in winter than those ($303 \pm 66$ mg C m$^{-2}$ d$^{-1}$) recorded during the summer. The overall averaged GP was 220 g C m$^{-2}$ year$^{-1}$ and NP was -3 g m$^{-2}$ year$^{-1}$, implying the reservoir was net heterotrophic on annual basis. This is due to negative NP rates from October to February ($-198 \pm 78$ mg C m$^{-2}$ d$^{-1}$). Comparisons between GP rates obtained from the isotope mass balance approach and $^{14}$C bottle incubation method ($^{14}$C-GP) showed consistent values on the same order of magnitude with a GP / $^{14}$C-GP ratio of $1.2 \pm 1.1$. Finally we noted that although typhoon occurrences were scarce, higher than average $^{17}\Delta$ values and GP rates were recorded after typhoon events.

**Key words.** Triple oxygen isotope anomaly, $^{17}$O-excess, oxygen : argon ratio, $^{14}$C bottle incubations, carbon cycle, primary production, gross production, net production, freshwater system, Feitsui Reservoir

## 1 Introduction

It is well established that marine photosynthesis plays a critical role in the global biogeochemical cycling of carbon and oxygen that sustain the great majority of ecosystems on our planet. Recent studies show that freshwater systems constitute a significant component of these cycles and deserve closer attention (Cole et al. 2007, Tranvik et al. 2009, Valdespino-Castillo et al. 2013). Assessing primary production (PP) and providing accurate estimates of ecosystem metabolic rates are therefore a key for understanding each system's fluxes and variability in biogeochemical cycling.

Traditionally, PP has been evaluated by in vitro $^{14}$C bottle incubation method introduced by Steeman-Nielsen (1952). However, these measurements are associated with a number of biases and the interpretation of the PP estimates is problematic. The main drawback is the in vitro methodology, which involves the removal of plankton communities from the natural environment and confining them into a small volume of water, with variability in PP observed under laboratory conditions. Because the distribution of plankton is heterogeneous in time and space, these experiments can only provide local and instantaneous PP rates, which do not reflect the time-averaged mean PP. The PP rates observed in vitro therefore can not be fully representative of natural PP rates (e.g. Harrison and Harris 1986, Marra 2002).

Over a decade and half ago, Luz et al. (1999) and Luz and Barkan (2000) introduced the triple oxygen-isotopes technique or the $^{17}$O-excess ($^{17}\Delta$), which allows to assess PP in situ. The excess is defined as

$$^{17}\Delta = ln(1 + \delta^{17}O) - \lambda \times ln(1 + \delta^{18}O) \tag{1}$$

where the isotopic compositions $\delta^{17}O$ and $\delta^{18}O$ represent the deviation of the abundance ratio of an isotopic and normal species in a sample relative to that of a standard: $\delta^*O = ([^*O]/[^{16}O])_{sample}/([^*O]/[^{16}O])_{standard}-1]$, $^*O$ is either $^{17}O$ or $^{18}O$. Here, $\delta^{17}O$ and $\delta^{18}O$ are expressed with respect to atmospheric air $O_2$. Following Luz and Barkan (2005), the factor $\lambda$ is taken to be 0.518. The basic premise of this method lies in the processes fractionating $O_2$ isotopologues. While photochemical reactions in the stratosphere (the coupled chemistry between $O_2$, $O_3$, and $CO_2$) give rise to a non-mass-dependent signal in the atmospheric $O_2$ (Luz et al. 1999), respiration and photosynthesis fractionate $O_2$ in a mass-dependent way (the $^{17}O$ enrichment is approximately half of the $^{18}O$ relative to $^{16}O$), which in a marine or aquatic systems allows for distinguishing the $O_2$ produced biologically from air $O_2$ entraining during gas exchange. Respiration modifies the dissolved $O_2$ concentrations in water but does not affect the $^{17}\Delta$, because the relative proportions of $\delta^{17}O$ and $\delta^{18}O$ remain the same. The respiratory effect on dissolved $O_2$ saturation can be evaluated using oxygen/argon ratios, considering the biological oxygen supersaturation expressed as $\Delta(O_2/Ar)$, (defined below in Eq. 4). This is because $O_2$ and Ar have similar physical properties, but the latter does not have biological sources and sinks. Although the PP evaluation based on the co-variation of both the $\delta^{17}O$ and the

$\delta^{18}O$ values provides a more accurate assessment (Prokopenko et al. 2011, Kaiser 2011), the concept of $^{17}\Delta$ remains a valuable tool for tracing biologically produced $O_2$ and resolving the associated dynamics in the ocean. Until now this joint geochemical budget approach ($^{17}\Delta$ or $\delta^{17}O$ and $\delta^{18}O$, together with $\Delta O_2/Ar$) has been applied widely to study marine production in the Atlantic (Luz and Barkan 2009, Quay et al. 2012), Pacific (Hendricks et al. 2005, Sarma et al. 2005, 2006, 2008, Quay et al. 2010, Prokopenko et al. 2011, Stanley et al. 2010, Juranek et al 2010, 2012, Munro et al. 2013) and the Southern Ocean (Reuer et al. 2007, Hamme et al. 2012, Huang et al. 2012, Castro-Morales et al. 2013), yet other oceanic basins and freshwater systems in general, with the exception of a case study in Lake Kinneret (Luz and Barkan 2000) remain by far and largely unstudied.

In this study, we extend the applicability of the $^{17}\Delta$ method to an aquatic system. We use the $^{17}\Delta$ method to trace the photosynthetic $O_2$ fate and to investigate the seasonal changes in PP in a semi-closed subtropical reservoir in Taiwan over a period of one year. We demonstrate that this approach offers new perspectives on PP in lakes. In an effort to contribute to the understanding of production rates measured in situ using the $^{17}\Delta$ method and the in vitro estimates from the $^{14}C$ bottle incubation approach, and to expand this to freshwater systems, we provide comparisons between the respective rates. Additionaly, we show data on the isotopic composition ($\delta^{18}O$, $\delta D$, and $^{17}\Delta$) of water from the reservoir. Understanding the isotopic composition of the Feitsui Reservoir water is crucial for accurate assessments of production rates using the $^{17}\Delta$ method and also offers insights into the biogeochemical/hydrological cycling of the reservoir. Ultimately, this paper presents a contribution to the studies on Feitsui Resevoir, a socio-economically and ecologically important reservoir.

## 2 Methods

### 2.1 Site description

The subtropical Feitsui Reservoir, located in northern Taiwan, is the country's second largest reservoir by volume (first is Tsengwen Dam in the south), serving as the main water source for over five millions of people in the Taipei metropolitan area. The domestic demand is supplied by water releases from the Feitsui Reservoir and unregulated flow from Nanshin Creek downstream of the watershed. The upstream watershed encompasses the Beishi stream basin a branch of Xindian River, one of the three major tributaries of Tamsui River. The total catchment area of the reservoir is 303 km$^2$ and storage volume at normal maximum water level is 406 million m$^3$. The mean depth of the watershed is 40 m with maximum depth of 113 m near the dam site. The mean daily inflow to Feitsui Reservoir is ~30 m$^3$/s and the amount of water released depends on the reservoir's storage capacity and whether flow from Nanshin Creek is sufficient to supply domestic demand (Shiau and Wu 2010). In the past the reservoir was found to alternate between mesotrophic and oligotrophic states (Kuo et al. 2003), although more recent studies (Kuo et al. 2006) observed a trend towards eutrophication. In 2012 according to Carlson's

Trophic State Index (CTSI) the reservoir was in a mesotrophic state. In order to prevent deterioration of water quality, the watershed is protected by the Feitsui Reservoir Administration with restricted access to the water as well as adjacent areas and any commercial and recreational activities are prohibited. In addition, since January 1988 the Feitsui Reservoir Administration operates a meteorological station that provided direct wind speed measurements 10 m above the water level and rainfall data used in this study. Typhoon information was obtained from a typhoon database (http://rdc28.cwb.gov.tw/TDB/ctrl_typhoon_range_search).

## 2.2 Physical structure and mixing in the water column of Feitsui Reservoir

Current assessments of PP rates rely on steady-state assumption. Whether the approximation is valid requires careful assessment and can be verified by studying the physical structure of the water body. Feitsui Reservoir is a typical monomictic system (characteristic of subtropical lakes), that stays thermally stratified throughout the greater part of the year, with changing intensity of winter vertical mixing depending on the meteorological conditions. The topographic characteristics of the reservoir, a large water mass located in a valley make its physical structure (i.e., water temperature) fairly simple and stable over the seasonal scale (Itoh et al. 2015). The residence time of water in the reservoir tends to be rather long; throughout our study we estimated it to be about 150 days, comparable to durations reported in the past (150 days reported by Kuo et al. 2003 and 115 days reported by Chen et al. 2006), sufficiently long to mix horizontally well. Field measurements as well as model simulation by Kuo et al. (2003) reported alike trends in dissolved oxygen concentration throughout a period of 12 months, recorded at the Dam Site (S1) and at the Wu-Tan station situated on the other side of the reservoir upstream of the Beishi Creek. A comparison between the top, middle, and bottom layers of the water column showed that the reservoir is horizontally rather uniform and is not affected significantly by horizontal water advection (Kuo et al. 2003).

## 2.3 Water sampling and sample preparation

Sampling was carried out at station S1 (24.9 E, 121.566667 N, Fig. 1) in the Feitsui Reservoir in the upper 100 m, located in the deepest region of the lake (~113 m). Water samples were collected using 5-L GO-FLO samplers with a manual messenger. For dissolved oxygen analysis, we collected waters at 9 depths (1, 5, 10, 15, 20, 30, 50, 70 and 90 m) during 13 separate trips to the reservoir, covering one full year from June 2014 to July 2015. Sampling for isotope analysis of water started later in September 2014.

Vertical profiles of temperature, chlorophyll *a,* and dissolved oxygen concentration were recorded routinely using Ocean Seven 316 CTD (IDRONAUT, Italy) multiparameter probe. A PAR sensor (BioTech) was used to measure photic irradiance. The casts were typically carried out on weekly basis during the summer and every two weeks during the winter. The

accuracy of dissolved oxygen measurements was verified against in vitro measurements; water samples collected from 10 depths (0, 2, 5, 10, 15, 20, 30, 50, 70 and 90 m) were siphoned intro triplicate 60 ml bottles (Wheaton) and a colorimetric method of Pai et al. (1993) was adopted for in vitro dissolved $O_2$ determination with precision of 0.2 % r.s.d. (full scale). We used a conservative approach for determining the mixed layer depth based on visual inspections of vertical temperature and dissolved oxygen profiles, to ensure the well-mixed layer only is described, without influences from the thermocline. For visualization and analysis of the profile data, we used Ocean Data View (ODV, Schlitzer 2015).

Dissolved gasses were extracted from water following Emerson et al. (1995) and Luz et al. (2002). In summary, 300 ml flasks with LouwersHapert© O-ring stopcock, containing 50 µl of saturated $HgCl_2$ solution, were evacuated prior to sampling and closed with a water lock. Approximately 150 ml of water sample was collected in the flask, leaving 150 ml of headspace for gases to exsolve. Once stopcock closed, the port was filled with the same water as sampled and sealed with a rubber cap to avoid air contamination. All samples were equilibrated for 24 h in a shaker at room temperature. After equilibration, water was removed from the samples and the flasks were subsequently connected to a preparation system for removal of water vapour, $CO_2$, and other condensable molecules at liquid nitrogen temperature. The extracted gases were then either stored in a sealed glass tube or directly introduced to a GC system (Thermo Scientific TRACE Gas Chromatograph) for complete removal of $N_2$ after which only $O_2$ and Ar remained the main components in the gas mixture. The separation was done using a chromatographic column (3 m long, 1/8" SS tube, packed with molecular sieve 5A at mesh 60/100), modified from Barkan and Luz (2003). During the separation the chromatographic column was kept at room temperature, and the yielded oxygen-argon mixture was absorbed onto two pellets of molecular sieve (1.6 mm, 5A, manufactured by SUPELCO) for subsequent isotopic analysis, following Abe (2008) with slight modifications.

## 2.4 [14]C bottle incubations

In summary, water samples were incubated for approximately 2-3 hours, and a chlorophyll-normalized photosynthesis rate versus light intensity (i.e. $P^B$ vs. I) model without the photoinhibition term proposed by Jassby and Platt (1976) was used to calculate primary production over 24 hours performed using an artificial light source to mimic the solar spectrum at an intensity controlled according to the solar irradiance measured in situ. The [14]C rates reflect gross C production and are integrated for the entire euphotic zone. Detailed description of methodology for [14]C incubation experiments is provided in Shiah et al. (1996).

## 2.5 Stable isotope analysis of dissolved oxygen

$\delta^{17}O$ and $\delta^{18}O$ in $O_2$ from the purified oxygen-argon mixture (as explained in Sec. 2.3) were determined by dual inlet mass spectrometry (Thermo Scientific Finnigan MAT 253 Stable Isotope Ratio Mass Spectrometer). Each sample was run for 3

acquisitions, 12 changeover cycles each, thus the reported δ values represent the average of 36 cycles. The analytical errors (1-σ standard error of the mean n=36 multiplied by Student's t-factor for a 95 % confidence limits, reported following Barkan and Luz, 2003) for $\delta^{17}O$ and $\delta^{18}O$ were 0.013 ‰ and 0.006 ‰, respectively. Our actual and long-term precision (1-σ standard deviation) established from routine measurements (n = 36) of atmospheric $O_2$ for $\delta^{17}O$, $\delta^{18}O$, and $^{17}\Delta$ was 0.017 ‰,

0.030 ‰, and 6 per meg, respectively (see the Online supplementary material, Table S1). Our current $O_2$ scale, reported also in Liang and Mahata (2015), is in agreement with that of Barkan and Luz (2011).

The $O_2$/Ar ratio was obtained by peak jumping; a sequential measurement of m/z '32' and '40' in the same collector (with the idle and integration times 20 and 4 s, respectively, following Barkan and Luz 2003) prior to isotopic ratio analysis. The

$O_2$/Ar ratio is expressed in the standard δ-notations and calculated as $\delta O_2/Ar$ (‰) = $[(32/40)_{sample}/(32/40)_{standard}-1]10^3$. The long-term precision (1-σ standard deviation) of routine measurements of atmospheric air was better than 5 ‰. For all water samples the final $\delta O_2/Ar$ values were corrected for the distribution of gases between the headspace and water in the sampling flasks, following Luz et al. (2002) and normalized to air. To verify the purity of the collected oxygen-argon mixture after the GC separation, we also included regular monitoring of m/z '28' signal during peak jumping. We did not detect any

significant presence of $N_2$ during the course of the study, either in atmospheric air or water samples for dissolved oxygen analysis. For all analysed samples the $N_2/O_2$ signal ratio was lower than 0.001.

Similar to Luz and Barkan (2003) and Abe and Yoshida (2003), we found that the values of $\delta^{17}O$, $\delta^{18}O$ and $^{17}\Delta$ are affected by the $O_2$/Ar ratio, presumably due to interference with the ion source of the mass spectrometer. Although this effect on the

$\delta^{17}O$ and the $\delta^{18}O$ values is minor, it may significantly affect the $^{17}\Delta$. We calculated the dependencies of the $\delta^{17}O$, $\delta^{18}O$ and $^{17}\Delta$ on the $O_2$/Ar ratio, derived from measurements of aliquots of pure $O_2$ with added different amounts of Ar and applied the correction to the reported final isotopic values. The regression slopes for $\delta^{17}O$, $\delta^{18}O$ and $^{17}\Delta$ were 0.00001 ‰ / ‰ ($R^2$ = 0.66), -0.00002 ‰ / ‰ ($R^2$ = 0.79), and 0.0217 per meg / ‰ ($R^2$ = 0.99), respectively.

To minimize the influence of Ar and for obtaining more precise results, we used a working $O_2$-Ar reference mixture from pure gases (>99.999 %) with the proportion of $O_2$/Ar ~20:1, similar to the $O_2$-Ar solubility ratio in surface water (Benson and Krause 1984, Krause and Benson 1989, Barkan and Luz 2003). The integrity of the standard was checked regularly by measuring aliquots of atmospheric $O_2$. For every set of samples for dissolved $O_2$ analysis from Feitsui Reservoir (one set representing one trip to the reservoir) three aliquots of atmospheric $O_2$ were prepared and measured against the same aliquot

of working reference gas mixture as used for the water sample set.

To evaluate the reproducibility and performance of sample preparation, we prepared air-equilibrated water. The equilibration was achieved by continuous stirring of 8 L of deionised water with added $HgCl_2$ in a circulator with temperature control at 25 °C over a period of 72 hours. Dissolved gases were extracted following the same procedure as applied for the reservoir samples (see Sec. 2.2). The reproducibility (1-σ standard deviation) for the analysis of equilibrated water samples (n = 3) was 0.020 ‰, 0.037 ‰, and 3 per meg for $\delta^{17}O$, $\delta^{18}O$, and $^{17}\Delta$, respectively and 4.6 ‰ for $\delta O_2/Ar$ (Table 1).

## 2.6 Stable isotope analysis of water

To identify the source of water in the reservoir, we carried out additional analyses of δD and $\delta^{18}O$ in the $H_2O$ molecule of reservoir water. For this, water samples were collected in 15 ml centrifuged vials and sealed with Parafilm M® to prevent any isotopic alteration due to evaporation. Prior to analysis, water was transferred to 2 ml vials with the aid of a pipette and analysed in a Picarro L2130-I Isotopic $H_2O$ Analyser, following Laskar et al. (2014). δD and $\delta^{18}O$ values are expressed with respect to VSMOW (‰). The $^{17}\Delta$ was determined using $CoF_3$ fluorination method following Barkan and Luz (2005). Briefly, an aliquot of 5 µl of water was converted to $O_2$ by injecting it to a $CoF_3$ containing reaction tube heated at 370 °C under helium flow. The evolved oxygen gas was collected in a 13X molecular sieve U-trap at liquid nitrogen temperature, and then determined by dual-inlet mass spectrometry (Thermo Scientific Delta Plus). Each sample was run for 80 changeover valve cycles, i.e., 80 sample-standard combinations. Mean standard deviations (1-σ) of multiple duplicate analyses for various waters (including VSMOW2, GISP, and SLAP) for $\delta^{17}O$, $\delta^{18}O$, and $^{17}\Delta$, were 0.086 ‰, 0.168 ‰, and 11 per meg, respectively.

## 2.7 Gross and net production calculations

Aquatic primary production (PP), the synthesis of organic compounds from aqueous carbon-containing species, in a steady state system may be distinguished as gross production (GP) and net production (NP). The GP represents the total carbon fixed by primary producers, and the NP represents the carbon available to the heterotrophic community. The NP is therefore the difference between GP and community respiration and corresponds to the overall metabolic balance of an ecosystem. NP can be positive or negative. NP is positive when GP exceeds respiration and the ecosystem may export or store organic C. The value is negative when respiration exceeds GP and the ecosystem respires more organic C than was able to produce. Both GP and NP are terms of fundamental interest in carbon cycle studies.

To quantify gross production rates from $^{17}\Delta$ values, a simple box model may be applied; mixed layer gross oxygen production (GOP) is assumed at steady state with respect to $^{17}\Delta$ and $O_2$ concentrations, and vertical mixing is neglected, following Luz and Barkan (2000).

$$GOP = K\,C_o\,(^{17}\Delta - {}^{17}\Delta_{eq})/({}^{17}\Delta_{bio} - {}^{17}\Delta) \tag{2}$$

where $C_o$ is the $O_2$ concentration at saturation using solubility coefficients from Benson and Krause (1984) and K is piston velocity (the coefficient for gas-exchange; Crusius et al. 2003, Wanninkhof et al. 2009). Here, the $^{17}\Delta_{eq}$ is the air-water

equilibrium, deviating from zero due to isotopic fractionation during $O_2$ invasion and the $^{17}\Delta_{bio}$ represents the value of purely biologically produced $O_2$. We calculated K (the piston velocity) from daily wind speeds at Feitsui Reservoir according to Wanninkhof et al. (1987) and Vachon and Prairie (2013), based on studies of gas transfer velocities in lakes of comparable sizes to Feitsui Reservoir. Because the gas concentrations in the mixed layer depend on the recent history of wind speeds, we averaged K over the residence time of $O_2$ in the mixed layer preceding sampling, based on the mixed layer depth and gas

transfer coefficient.

Equation 2 however represents a mathematical approximation, to provide a first order realization of processes and sources that affect the GP. This simplified formulation may introduce large errors, in particular in ecosystems with elevated export ratios. Prokopenko et al. (2011) and Kaiser (2011) derived an improved 'dual-delta approach', which we applied for

estimating GP rates in the Feitsui Reservoir, where the GOP may be directly calculated from the measured $\delta^{17}O$ and $\delta^{18}O$ values, as follows:

$$GOP = K\,C_o\left[\frac{\left(1-\frac{1+\delta^{17}O_{eq}}{1+\delta^{17}O}\right)-0.518\left(1-\frac{1+\delta^{18}O_{eq}}{1+\delta^{18}O}\right)}{\left(\frac{1+\delta^{17}O_p}{1+\delta^{17}O}-1\right)-0.518\left(\frac{1+\delta^{18}O_p}{1+\delta^{18}O}-1\right)}\right] \tag{3}$$

where $\delta^*O$ is the measured value in a sample, $\delta^*O_{eq}$ is the air-water equilibrium, and $\delta^*O_p$ represents the photosynthetic $O_2$.

To estimate the net oxygen production (NOP) rates, we have used the $O_2/Ar$ measurements, following the biological $O_2$ supersaturation concept for net photosynthetic production. Because the physical properties of $O_2$ and Ar are similar, and Ar has no biological sources and sinks, measurements of Ar concentration in water may be used to remove physical

contributions to $O_2$ supersaturation. The biological oxygen supersaturation $\Delta(O_2/Ar)$ is defined as the relative deviation of the $O_2/Ar$ in a sample to the $O_2/Ar$ at equilibrium with the atmosphere (Craig and Hayward 1987, Spitzer and Jenkins 1989, Emertson et al. 1995, Kaiser et al. 2005) and may be calculated as follows:

$$\Delta(O_2/Ar) = \left[\frac{1+(\delta O_2/Ar)_{sample}}{1+(\delta O_2/Ar)_{eq}} - 1\right] \tag{4}$$

Assuming the mixed layer is at steady state, NOP can be calculated following Luz et al. (2002):

$$NOP = K\,C_o\,\Delta(O_2/Ar) \tag{5}$$

A shortcoming associated with the calculation of PP rates from dissolved $O_2$ isotopes is that the rates are in $O_2$ units, instead of C based units, and the conversion between them is not straightforward. To convert between $O_2$ and C based rates, we follow the common approach presented earlier (e.g., Hendricks et al. 2014, Juranek et al. 2012). GOP from $^{17}\Delta$ is greater than gross C production because it measures total oxygen produced regardless of its fate, such as the fraction of $O_2$ produced which is linked to Mehler reaction and photorespiration. To scale GOP to gross C production, we account for this fraction following Laws et al. (2000) and apply a photosynthetic quotient (PQ) of 1.2. We convert NOP to a comparable C flux using a PQ of 1.4, for new production (Laws 1991).

## 3 Results

### 3.1 Hydrography

We refer to our monthly sampling dates as MMMYY for convenience. The subtropical Feitsui Reservoir was thermally stratified for the great part of the year, with a distinct seasonal thermocline (Fig. 2a). In spring seasonal stratification developed (APR15), and the lake remained well stratified with a shallow epilimnion with temperature above ~30 °C throughout the warmer months in the top 10 m layer. In summer, the reservoir was strongly stratified as the result of continued heating of the surface water and the mixed layer remained shallow, typically about 3 – 5 m deep, as observed in JUN14, AUG14, APR15, MAY15 and JUL15. Although the thermal structure of the reservoir controls the gas exchange during the warm months, processes such as rainfall and windstorms may entrain atmospheric air to the thermocline, as showed from dissolved $O_2$ and $\Delta(O_2/Ar)$ vertical profiles (Fig. 2c and 4c), playing a critical role in influencing the conditions in the mixed layer. From SEP14 and OCT14, as a result of decreasing ambient temperature and gradual cooling of surface water the mixed layer deepened reaching 11 and 23 m, respectively. In DEC14 the thermal stratification became weaker initiating the winter overnturn, resulting in well-mixed epilimnion of ~22 °C to 34 m depth, and a decreasing temperature gradient in the metalimnion. In JAN15 the mixed layer was deepest at 51 m. The mixed layer remained deep throughout FEB15 (40 m) when the reservoir was coolest and nearly homothermal, ~17 °C in the epilimnion and ~16°C in the hypolimnion.

The Chlorophyll *a* (Chl *a*) fluorescence (Fig. 2b) in the reservoir was predominantly restricted to the epilimnion and the upper thermocline. The distribution and concentration varied with seasons and with the occurrence of stochastic events (such

as storms, strong rainfall or typhoons) that enhanced the photosynthetic activity in the phytoplankton. Chl $a$ concentration was high from JUN14 to SEP14, with a subsurface maximum below the mixed layer at ~10 m, averaging to 15 mg m$^{-3}$ and occasionally above 20 mg m$^{-3}$. In late SEP14, the Chl $a$ maximum shifted to the surface. No apparent maximum was observed in OCT14 when Chl $a$ concentration was rather uniform throughout the mixed layer of average ~8 mg m$^{-3}$ in the upper 23 m. From DEC14 to FEB15, Chl $a$ remained low at ~3 mg m$^{-3}$, with the exception of a small episodic subsurface maximum of ~5 mg m$^{-3}$ at 8 m in JAN15. In spring, Chl $a$ concentration started to increase at the surface averaging ~5 mg m$^{-3}$, followed by later appearance of a subsurface maximum at 12 m of ~8 mg m$^{-3}$. A short episodic decrease in Chl $a$ was recorded in APR15, a likely result of cooler ambient temperature than normal and frequent precipitation. In MAY15, Chl $a$ concentration increased again and high subsurface maximum of >10 mg m$^{-3}$ formed at 15 m.

The dissolved oxygen (DO) concentration and saturation levels (Fig. 2c) varied in association with $O_2$-Ar ratios measured, indicating the availability of dissolved $O_2$ supplied by primary production, aeration or mixing processes. The epilimnion remained saturated throughout the year, in the thermocline DO varied greatly; DO was undersaturated from AUG14 to DEC14 (<50 %), reached near-saturation levels during the winter and from APR15 to JUN15 remained supersaturated (>100 %). In spring and summer the DO reached the hypolimnion where the saturation was typically above 50%. Conversely, from early AUG14, throughout autumn and winter the hypolimnion was undersaturated (< 50 %) with minimal DO content.

### 3.2 Isotopic composition of water

In addition to dissolved $O_2$, we have measured the isotopic composition of water in the Feitsui Reservoir throughout the different seasons. The isotopic composition of water varied in both $\delta^{18}O$ (Fig. 3a) and $\delta D$ (Fig. 3b) seasonally and vertically. Overall, the variation of $\delta^{18}O$ was smaller than that of $\delta D$, varying between -6.5 and -5.3 ‰ in $\delta^{18}O$ and between -37.1 and -25.4 ‰ in $\delta D$. The general pattern for both, $\delta D$ and $\delta^{18}O$, showed more depleted values during autumn, followed by gradual enrichment throughout the winter, spring and early summer. No statistically significant (within errors) seasonal variation was found in $^{17}\Delta$ values. We therefore averaged the $^{17}\Delta$ over the year, with the resultant $^{17}\Delta$ of 257 ± 14 per meg, representing the $^{17}\Delta$ of water in the Feitsui Reservoir. The insignificance in $^{17}\Delta$ seasonality is consistent with the small overall variation in $\delta^{18}O$ and also with the long residence of water in the reservoir.

### 3.3 The $^{17}\Delta$ of dissolved $O_2$

The $^{17}\Delta$ signal of dissolved $O_2$ varied with depth and seasons (Fig. 4b). The overall range of the $^{17}\Delta$ values measured ranged between the maximum of 205 per meg in late AUG14 and JUL15 and minimum of 26 per meg in JUN14. The annual mean $^{17}\Delta$ at the surface (1 m) was 65 ± 13 per meg and remained constant throughout the year, with the exception of late SEP14 and JUL15, when the surface values were higher, recording 97 and 76 per meg, respectively and coinciding with an episodic

shift of Chl *a* maximum towards the surface (Fig. 2b). During months with persistent thermal stratification in the reservoir, the $^{17}\Delta$ followed a similar vertical pattern, with distinct $^{17}\Delta$ values. From JUN14 to early SEP14, when the mixed layer was very shallow (~3-5 m), the $^{17}\Delta$ signal accumulated below, in the upper thermocline, with a peak exceeding 150 per meg observed typically at ~10-20 m depth. Below the thermocline at ~50 m depth in AUG14 and SEP14 the $^{17}\Delta$ was low showing signals characteristic of surface water. In late SEP14 and OCT14 as the mixed layer deepened gradually, we observed a corresponding trend for the $^{17}\Delta$, with the peak in the signal deepening (205 and 156 per meg at ~20 and 30 m, respectively) following the mixed layer boundary. In DEC14 the deep mixed layer and decreasing importance of the thermal stratification facilitated increased gas exchange throughout the water column resulting in low and rather uniform $^{17}\Delta$ values in the water column of 63 ± 6 per meg. This trend continued throughout the winter period, during which the $^{17}\Delta$ values stayed comparatively low and less variable. The onset of thermal stratification during the spring allowed for $^{17}\Delta$ to increase below the mixed layer; in APR15 we observed a developing peak in the $^{17}\Delta$ signal of 105 per meg at 10 m. The $^{17}\Delta$ increased throughout MAY15, and in JUN15 and JUL15 the accumulated $^{17}\Delta$ signal at 10-30 m, as high as 193 per meg was seen, coinciding with the observed DO supersaturation (Fig. 2c) and elevated $\Delta(O_2/Ar)$ (Fig. 4c). In contrast to the trend from JUL14, in JUL15 the $^{17}\Delta$ signal was high below the mixed layer and throughout the whole water column. Although samples from regions below 50 m and 70 m were limited due to insufficient amount of gas for isotope analysis, overall the $^{17}\Delta$ values tend to increase towards the bottom of the hypolimnion, in particular high $^{17}\Delta$ was measured at 70 m in late AUG14, SEP14, and JUL15 of 152, 157 and 174 per meg, respectively.

## 3.4 Gross and net production

Following Prokopenko et al. (2011) and Kaiser (2011), we estimated PP rates using the 'dual delta' approach in the Feitsui Reservoir from JUN14 to JUL15. The GP rates, NP rates and the NP/GP ratio obtained by the dual-delta approach are summarised in Table 2. Overall, the GP rates varied between 187 and 1372 mg C m$^{-2}$ d$^{-1}$. The general pattern showed higher values during the cooler months and in winter, averaging 702 ± 107 mg C m$^{-2}$ d$^{-1}$ between OCT14 and APR15. This may be considered as maximum production because the mixed layer is deeper than the euphotic zone in winter. A decrease in the GP was observed in summer, averaging 306 ± 66 mg C m$^{-2}$ d$^{-1}$ throughout JUN14, AUG14, MAY15 and JUN15. This represents the minimum GP for this period, because the mixed layer is shallower than the euphotic zone and therefore some production also took place below the mixed layer, which may not be evaluated by the present model. Production was highest in late SEP14 and in JUL15, 1372 and 1162 mg C m$^{-2}$ d$^{-1}$, respectively, coinciding with typhoon events affecting the area of the Feitsui Reservoir.

Overall, the NP ranged between -311 and 228 mg C m$^{-2}$ d$^{-1}$. The NP was negative from OCT14 to FEB15, averaging -198 ± 78 mg C m$^{-2}$ d$^{-1}$, indicating the reservoir was net heterotrophic in the mixed layer during the winter. Positive NP rates

dominated in the warmer months with values typically averaging $57 \pm 26$ mg C m$^{-2}$ d$^{-1}$, implying the reservoir remained net autotrophic during the greater part of the year. Highest NP rates were observed in JUL15 measuring 228 mg C m$^{-2}$ d$^{-1}$. The NP / GP ratio varied between –0.44 and 0.30.

5    Using the GP and NP rates measured in our study, we estimated the annual C production in the Feitsui Reservoir. Excluding the measurements obtained during episodic typhoons (late SEP14 and JUL15), the average annual GP amounted to 177 g C m$^{-2}$ year$^{-1}$ and the average annual NP was -12 g C m$^{-2}$ year$^{-1}$. Taking into consideration typhoon events, the average annual GP increases to 220 g C m$^{-2}$ year$^{-1}$, and the average annual NP to -3 g C m$^{-2}$ year$^{-1}$. For comparison, a model study by Lewis (2011) estimates the global average annual production per unit area for a lake 200 g C m$^{-2}$ year$^{-1}$ for GP and 160 g C m$^{-2}$

10   year$^{-1}$ for NP. While our GP estimates agree well with this projection, in particularly when the production rates take into account typhoon events, our NP rates lie at the lower end of the global average. The measured NP rates in the Feitsui Reservoir thus indicate that over the year the respiration exceeds gross primary production in the reservoir, affecting the net balance of carbon. Episodic events such as typhoon events seem to play a key role in the metabolism of Feitsui reservoir, and it is plausible that during years with frequent typhoon events the annual balance of the reservoir may shift to net autotrophy.

Although some carbon storage is expected in the form of accumulation of organic matter to the sediments (Dean and Gorham 1988), our results indicate that from 2014 to 2015 Feitsui reservoir acted as a positive though minor carbon source to the atmosphere.

## 4 Discussion

### 4.1 The $^{17}\Delta$ and $\Delta(O_2/Ar)$ tracers for photosynthesis and respiration

The schematics of $^{17}\Delta$ transport and variation are summarised in Fig. 5. The near-surface $^{17}\Delta$ value represents a balance between O$_2$ produced photosynthetically, which tends to increase the $^{17}\Delta$, and that from gaseous exchange with atmospheric O$_2$, which reduces the $^{17}\Delta$ value. The nearly constant surface $^{17}\Delta$ values measured in the Feitsui Reservoir throughout the year ($65 \pm 13$ per meg) suggest that the balance of these processes does not typically vary with seasons. In late SEP14 and in JUL15, the surface $^{17}\Delta$ values were 32 and 10 per meg higher, respectively, than the annual mean, indicating additional input

from photosynthesis. Further analysis showed that in both cases, the samples were taken within a few days after typhoon occurrences. Thus the resulting elevated $^{17}\Delta$ is likely to be a consequence of nutrient enrichment caused by typhoons, mediating enhanced vertical mixing and hence photosynthesis (see Sect. 4.5 for further details). Previous studies showed that in the Feitsui Reservoir phosphate plays a key role as the limiting nutrient, restricting the microbial production. The key processes that determine its availability are vertical mixing from changes in the mixed layer depth in the spring and typhoon

intensity in summer and autumn (Tseng et al. 2010, Itoh et al. 2012). These processes therefore likely play a role in the

distribution of the $^{17}\Delta$ signal in the water column as well as increase photosynthetic activity as a result of intensified production after nutrient enrichment.

High subsurface $^{17}\Delta$ may be primarily attributed to the decreasing importance of gas exchange with depth. This is particularly characteristic of the warmer months during which strong thermal stratification developed confining the primary producers to the thermocline (also showed by Chl $a$, Fig. 2b) where the conditions are optimal for phytoplankton growth, representing a compromise between light, temperature, and nutrient availability. From AUG14 to OCT14 we recorded high $^{17}\Delta$ (often above 150 per meg) values below the thermocline (5–30 m). It is likely that the local primary production was initially high, a possible result of a phytoplankton bloom influencing the observed $^{17}\Delta$ composition. Yet, the measured low $\Delta(O_2/Ar)$ values (about -30 %) and undersaturation of DO indicate $O_2$ consumption from AUG14 to OCT14 in the upper thermocline. In the absence of photosynthesis, the residual $^{17}\Delta$ signal thus points towards the lack of vertical mixing in the reservoir during this period as well as no influence from atmospheric air. At 50 m depth, we recorded $^{17}\Delta$ values, typical of near-surface water. Here, the $^{17}\Delta$ shows an inverse relationship to $\Delta(O_2/Ar)$ as well as DO, with lower $^{17}\Delta$ and higher $\Delta(O_2/Ar)$ and DO observed in AUG14 and SEP14 and increasing $^{17}\Delta$ and decreasing $\Delta(O_2/Ar)$ DO saturation observed in OCT14. Additionally, the signal also follows the thermal structure of the reservoir; from July to about November 2014 we observe well mixed epilimnion in the upper ~10 to 20 m and an extensive metalimnion to about 50–60 m, with strong thermal gradient before reaching hypolimnion below. It is likely that the low $^{17}\Delta$ origins from atmospheric air entrainment during early summer (June and July, DO profile, Fig. 2c).

Intrusion of surface water below the metalimnion has also been observed in previous studies and may also be supported by dust loading which results in an increase in total suspended material at depth (Tseng et al. 2010). Strong vertical mixing of air-saturated water down the water column in JUN15, indicated by the low $^{17}\Delta$ values, increased $\Delta(O_2/Ar)$, as well as DO, as a result of heavy rainfall (~ 700 mm accumulated precipitations during JUN14, comparing to ~340 and ~200 mm measured in JUL14 and AUG14, respectively, Fig. 6) may have supplied atmospheric $O_2$ to the metalimnion and the hypolimnion. As of early AUG14 at 50 m, the observed $^{17}\Delta$ signal possibly traces the remaining $^{17}\Delta$ signal from JUN15, locally confined due to the strong thermal gradient and unaltered by photosynthesis due to the lack of primary producers at this depth. The breaking down of both, the high $^{17}\Delta$ from ~20–30 m, and the low $^{17}\Delta$ from 50 m, throughout autumn is controlled by decreasing air temperatures and the consequent weakening thermal stratification in the reservoir, also observed in other years (Itoh et al. 2012). Apart from storms and typhoons causing wind stress at the air-water interface or heavy precipitation, other processes that affect vertical mixing in the reservoir include instabilities caused by heat losses at the surface or as a result of lake processes such as seiches. Seiches were never evaluated in the Feitsui Reservoir, but they may play an important role in

affecting the vertical transfer of the water masses, dissolved gases, and nutrients in the Feitsui Reservoir and should be considered in future studies.

Increase in the $^{17}\Delta$ towards the bottom of the lake (90 m samples) was observed during all seasons, originating likely from the transport of enhanced $^{17}\Delta$ values from the upper part of the water column and any photosyntheticallty induced changes to

the signal before it reached the bottom. These samples however contained only small amounts of $O_2$ (saturation less than ~50 %), and therefore it is possible that minor photosynthetic contributions could significantly increase the $^{17}\Delta$ values, as a result of vertical entrainment mentioned previously.

## 4.2 Uncertainties in PP rates

Although the improved dual-delta method presents a mathematically more accurate approximation than the previous $^{17}\Delta$

model, a number of uncertainties associated with both methods for estimating PP rates remain, which need closer attention. Luz and Barkan (2000), Prokopeno et al. (2011), and Kaiser (2011) demonstrated that GP in the mixed layer could be determined from the measurements of $^{17}\Delta$ or the $\delta$-values in dissolved $O_2$ using a steady state mixed layer oxygen budget model which allows for estimation of integrated gross productivity in the mixed layer over the residence time of mixed-layer $O_2$. It is important to note that this approach may underestimate GP on occasions when the euphotic zone is deeper than the

mixed layer since the calculation accounts for GP in the mixed layer only. This may particularly affect PP estimates during summer months, when the photic layer is typically deeper than the mixed layer in subtropical reservoirs in general, and about 4 times deeper in the Feitsui reservoir. Furthermore, this model lacks terms for advection and vertical mixing. While the effect of these simplifications may be negligible in the open ocean (Emerson et al. 1997), lakes and reservoirs often feature a complex vertical and horizontal structure, the effect of which needs to be considered. Fortunately, given the rather simple

physical structure of the Feitsui Reservoir (see Sect. 2.2) horizontal inhomogenities may be neglected. The column inventory approach presented below (Sec. 4.3) provides a robust technique for assessing the contributions of vertical mixing and to estimate the production below the mixed layer.

The key parameter to constrain the GP and NP rates is the gas exchange rate between the mixed layer and the atmosphere.

Presently this is best achieved by parameterization of wind speeds, which is commonly used in models with several empirical relationships between the wind speed and gas exchange rate (e.g. Clark et al. 1995, Ho et al. 2006, Wanninkhof et al. 2009). Yet parameterization of wind speeds does not come without inaccuracies. While in most of the oceanic studies, the error associated with the parameterization is attributed to the accuracy of wind speed measurements and the relationship between the wind speed and gas exchange rate at very high or low wind speed conditions (Wanninkhof 1992). In freshwater

systems, factors such as lake size and ecosystem heterogeneity present another important factor (Vachon and Prairie 2013) and should be taken into consideration when choosing an appropriate parameterization.

Apart from δ-values measured in samples, $\delta^*O_{eq}$ and $\delta^*O_p$ or $^{17}\Delta_{eq}$ and $^{17}\Delta_{bio}$ are important constituents of this method. $\delta^*O_{eq}$ or $^{17}\Delta_{eq}$ is rather well established and can be determined experimentally by air-water equilibrations, usually achieved by bubbling or stirring (Keedakkadan et al. 2015). The $\delta^*O_p$ or $^{17}\Delta_{bio}$ represents the photosynthetic $O_2$ composition, with controversy on these values discussed in the literature (e.g. Kaiser 2011, Luz and Barkan 2011, Nicholson 2011). The major problem lies in proper quantification of the value for $^{17}\Delta_{bio}$. Being closely dependent on the $^{17}\Delta$ of substrate water and less straightforward to measure, the $^{17}\Delta_{bio}$ value was previously often assumed to be the same that of water. Recently, Luz and Barkan (2011) showed that a small difference exists between the substrate water values and the average composition of photosynthetic $O_2$ produced by phytoplankton. This potential bias has to be considered, in order to improve the accuracy of primary production estimates. Furthermore, whereas a uniform value for $^{17}\Delta$ of seawater may be applied to study PP in the ocean, $^{17}\Delta$ for a freshwater system of interest has to be determined because the isotopic composition of freshwater tends to vary geographically and among different water sources (Luz and Barkan 2010). For Feitsui Reservoir we determined 257 ± 14 per meg for $^{17}\Delta$ of water, based on measurements of water samples collected throughout the year. The difference between the isotopic composition of Feitsui Reservoir water and that of photosynthetic $O_2$ provided by Luz and Barkan (2011), therefore reflects the associated fractionation between the substrate water and the photosynthetic $O_2$. To obtain the representative $\delta^*O_p$ and $^{17}\Delta_{bio}$ values for the Feitsui Reservoir, we consider these additional fractionations of 26 per meg and 3.306 ‰ for $^{17}\Delta_{bio}$ and $\delta^{18}O$, respectively, to our measured values of water and retrospectively calculate the $\delta^{17}O_P$. The resulting annual mean values for $\delta^{17}O_P$, $\delta^{18}O_P$ and $^{17}\Delta_{bio}$ were -13.156 ± 0.192 ‰, -25.975 ± 0.370 ‰, and 283 ± 9 per meg, respectively.

Compared to the GP, estimating the NP is less complicated, since the model only requires the coefficient for gas exchange and a term describing the biological supersaturation. The second term can be constrained by the $\Delta(O_2/Ar)$ based on measurements from flask samples or determined in situ using a sensor for dissolved $O_2$ supersaturation, although the accuracy of the latter is inferior and may be less suitable for this purpose. Combining $^{17}\Delta$ and $\Delta(O_2/Ar)$ measurements we can get the NP/ GP ratio, which is equivalent to an export ratio (Laws et al. 2000) describing the capacity of an ecosystem to export C. The NP/GP ratio can be far better constrained than the GP on its own, since it is independent of the gas exchange rate and the uncertainty in the ratio only depends on the error in the measurement of $^{17}\Delta$ and $\Delta(O_2/Ar)$.

### 4.3 Column inventory approach

As discussed previously (Sec. 4.1), in case of fast changing physical dynamics in a reservoir, a mixed layer budget by isotope mass balance calculation may not be applicable for the assessment of PP rates. To assess the relevance of this method for the Feitsui Reservoir, we tested an alternative mass balance model based on a whole column inventory approach (onwards referred to as column inventory approach or CI-GP). Unlike isotope mass balance limited to the mixed layer, the column inventory model requires time-series data of full profiles from the surface to the bottom of the lake, and is able to obtain the GP rates below the mixed layer, without steady-state assumptions.

Calculating the GP by the column inventory model is done by solving the following simultaneous equations:

$$^{16}O_t - {^{16}O_{t-1}} = {^{16}P} - {^{16}C} + {^{16}I} - {^{16}E} \tag{Eq .6}$$

$$^{17}O_t - {^{17}O_{t-1}} = {^{17}P} - {^{17}C} + {^{17}I} - {^{17}E} \tag{Eq. 7}$$

$$^{18}O_t - {^{18}O_{t-1}} = {^{18}P} - {^{18}C} + {^{18}I} - {^{18}E} \tag{Eq. 8}$$

where $^{n}O_t$ and $^{n}O_{t-1}$ are the total amount of oxygen isotope n in the water column from the surface to the bottom of lake at the time slice t and t-1 (a step before time t), respectively; $^{n}P$, $^{n}C$, $^{n}I$ and $^{n}E$ are GP, consumption rate for entire water column, influx from the atmosphere and efflux to the atmosphere, respectively for oxygen isotope n. Equation (6) can be substituted by column inventory or rates of total dissolved oxygen Eq (9).

$$O_t - O_{t-1} = P - C + I - E \tag{Eq. 8}$$

Equations (7) and (8) can be obtained by multiplying isotopic composition $(1 + \delta^{n}O)$ and / or isotope fractionation factor $(1 + {^{n}\varepsilon})$ in the Eq (9)

While a non-steady state model is beyond the scope of this paper, the column inventory approach enables us to introduce dynamics into the calculations of GP and evaluate the feasibility of the mixed layer (ML) approach for estimating PP rates in the Feitsui Reservoir. Because the purpose of this model is to compare two different approaches, rather than produce accurate estimates, and for simplicity, we calculated the mixed layer approach GP rates using Eq (2) and present the $^{17}\Delta$-GP / CI-GP ratio (Fig. 7). Overall, the GP rates obtained from $^{17}\Delta$-GP and CI-GP model showed a good agreement with each other, indicating the calculation of GP rates using a mixed layer model may be valid for the Feitsui Reservoir, not only for open oceans. A better fit between the respective rates may be obtained using lambda slope 0.520, although the reasons for this remain presently unclear. Further adjustments of $^{17}\Delta_{eq}$ and $^{17}\Delta_{bio}$ could also improve the fit.

## 4.4 Comparisons between 'dual-delta' GP and [14]C-GP rates

While both methods aim to evaluate the natural GP rates, direct comparisons between estimates from the 'dual delta' method and from the [14]C bottle incubation are impractical because of the principal differences in the methodologies (in situ vs. in vitro). Each method provides rates integrated over different spatial and temporal scales, and clearly methodological biases are associated with each. A number of studies have addressed the 'dual delta' GP / [14]C-GP in the ocean, but the ratios were found to vary significantly from 2.2 (Quay et al. 2010) to 8.2 ± 4.0 (Stanley et al. 2010; see also Juranek and Quay 2013 for an extensive review). The variability in the ratios remains a conundrum. Overall, the 'dual delta' GP method showed a tendency to yield higher production rates. The factors responsible for the variability in the GP / [14]C-GP have however yet to be properly identified, before the gross $O_2$ production and carbon fixation can be properly linked and compared.

Contrasting the GP rates from 'dual delta' approach and the [14]C-GP rates (Table 2 and Fig. 6) we note that both estimates show a very similar trend. Moreover the production rates are on the same order of magnitude, with the average GP/[14]C-GP ratio of 1.2 ± 1.1. Although, due to the reasons mentioned above, it is unclear why the respective GP rates may agree or disagree, our results support the findings from an earlier study by Luz and Barkan (2000) who demonstrated near equivalence of gross production rates obtained from incubation-dependent and incubation-independent methods from Lake Kinneret. Presumably, near 1 GP / [14]C-GP ratios are characteristic of systems with shallow mixed-layer and rapid $O_2$ turnover, such as subtropical reservoirs in general, including the Feitsui Reservoir. The disparity between the GP and [14]C GP rates from AUG14 and early SEP14 could be explained by the shallow summer mixed layer, which would provide minimal GP rates due to the limitations of this approach. Conversely, it could also be that the [14]C-GP is overestimating the production rates particularly during this period, when algal blooms are more likely to occur. This highlights one of the key assets of the GP method, which in principle is not significantly affected by small-scale short-term events. From DEC14 onwards, the 'dual delta' GP rates and [14]C-GP show a close fit. Throughout the winter months, both methods showed rather invariable GP rates, though the 'dual delta' result were by approximately a factor of 2 higher than the [14]C-GP rates, a possible attribute of the integration on which each method operates. Alike, but reverse trend was also observed in MAY15, but onwards in JUN15 and JUL15 both methods showed a close fit.

## 4.5 Typhoon effects

Passing of tropical cyclones had been documented to cause entrainment and upwelling or 'atmospheric pumping,' injecting nutrients into the mixed layer which may significantly elevate PP. In the South China Sea, Lin et al. (2003) reported that the occurrence of only a moderate cyclone led to a 30-fold increase in the concentration of surface Chl *a*. Ko et al. (2015) studied the phytoplankton responses to typhoons in the Feitsui Reservoir and found a twofold increase in the phytoplankton level during typhoon periods. Conversely, in regions with deep mixed layer and nutricline, typhoon events may not be

sufficient to increase PP or induce phytoplankton blooms (Lin 2012). The effect of typhoon events on ecosystems is however complex and difficult to document properly because of their sporadic occurrence. Although our data are too limited to draw solid conclusions, we briefly discuss our results in context of typhoon events.

Two typhoons closely affected the north-eastern Taiwan and the Feitsui Reservoir during our study period. Typhoon Fung Wong hit Taiwan on the 22nd of September 2014 and typhoon Chan Hom on the 10th of July 2015, 1 and 4 days before the sample collection, respectively. Post-typhoon sampling occasions are indicated in Table 2. On both occasions, we found a considerable increase in the GP in the mixed layer (to 1372 and 1162 mg C $m^{-2}$ $d^{-1}$), representing approximately a 3-fold increase in the GP rates to those obtained on previous sampling days (Fig. 6), suggesting a critical role of typhoons in

modifying the seasonal metabolic balance of the reservoir. This corresponds to 32 and 10 per meg increase in surface $^{17}\Delta$. However, short episodic events of high production are normally expected to average out by the lower background $^{17}\Delta$ of the mixed layer due to elevated gas exchange with air. It is plausible that if the mixed layer is very shallow and the photosynthetic $O_2$ production is high, the elevated $^{17}\Delta$ signal would remain for an extended period of time. Alternatively, increased $^{17}\Delta$ values could arise from ventilation from water below the mixed layer or enhanced vertical mixing. Greater K

caused by higher wind speeds during a typhoon event could explain the higher GP rates; however, it does not explain the increase in $^{17}\Delta$ signal. Nevertheless it is important to note that these GP rates should be considered as minimum values, because on both occasions the thermocline was situated in the photic zone and therefore some of the production also took place below the mixed layer.

**5 Conclusions**

In summary, the $^{17}\Delta$ and $\Delta(O_2/Ar)$ values showed strong seasonal and vertical variations, enabling us to monitor the photosynthetic activity versus atmospheric $O_2$ input in the Feitsui reservoir. While our GP estimates situate the Feitsui Reservoir close to the global average, the low average annual NP was low indicated overall net heterotrophy in the reservoir in 2014-2015. The extent to which the reservoir acts as a carbon sink or source to the atmosphere is likely determined by typhoons, which may play a key role in enhancing seasonal mixing in the water column. Although the application of the

geochemical budget approach for PP evaluation in freshwater systems is less straightforward than in the ocean, and whether steady state assumptions may be valid needs prior assessments, it may offer us new perspectives on studying PP rates in situ. Even though we showed that the 'dual delta' GP and $^{14}$C-GP rates display to some extent a comparable trend, it is necessary to resolve the varying ranges in GP / $^{14}$C-GP observed between studies, and whether it reflects a real difference in the ratio of gross $O_2$ evolution to C fixation, or it should be attributed to methodological biases. Further studies addressing the questions

on various spatial and temporal scales will help us understand the full scope of the geochemical approach to PP evaluation,

in particular in dynamic and well-characterised environments that could serve as 'natural laboratories', such as the Feitsui Reservoir.

## 6 Data availability

The critical data used in this study are available in the Online supplementary material accompanying this contribution. CTD
and meteorological data are available upon request.

## Acknowledgements

We thank the Taipei Feitsui Reservoir Administration Bureau and the Environmental Ecosystem Laboratory group for assistance with fieldwork and making their data available to us. The authors would like to thank Sasadhar Mahata and Ho Wei Kang for their invaluable technical expertise and insights, and two anonymous reviewers whose comments helped to
10 significantly improve this manuscript. The work was supported in part by MOST grants 101-2628-M-001-001-MY4 and 105-2111-M-001-006-MY3 to Academia Sinica.

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

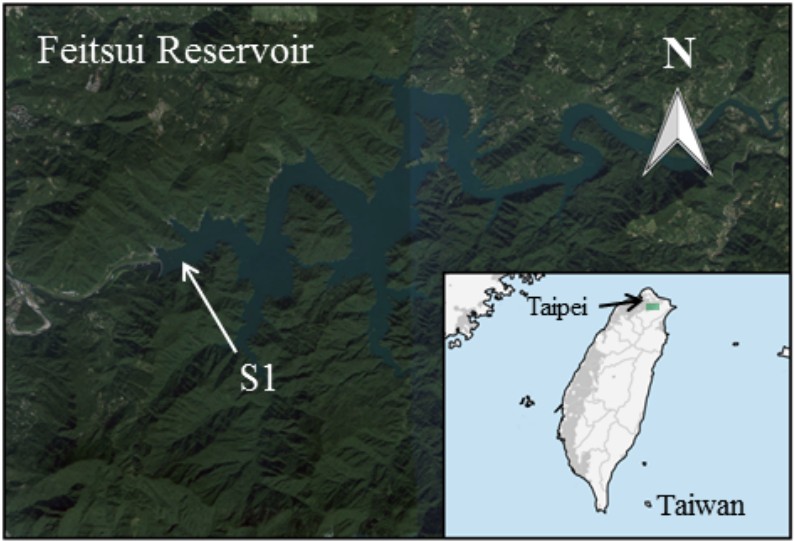

**Figure 1. Location of Feitsui Reservoir in northern Taiwan. Small green rectangle indicates the enlarged satellite map of the reservoir with the position of the long-term station S1 near the dam indicated.**

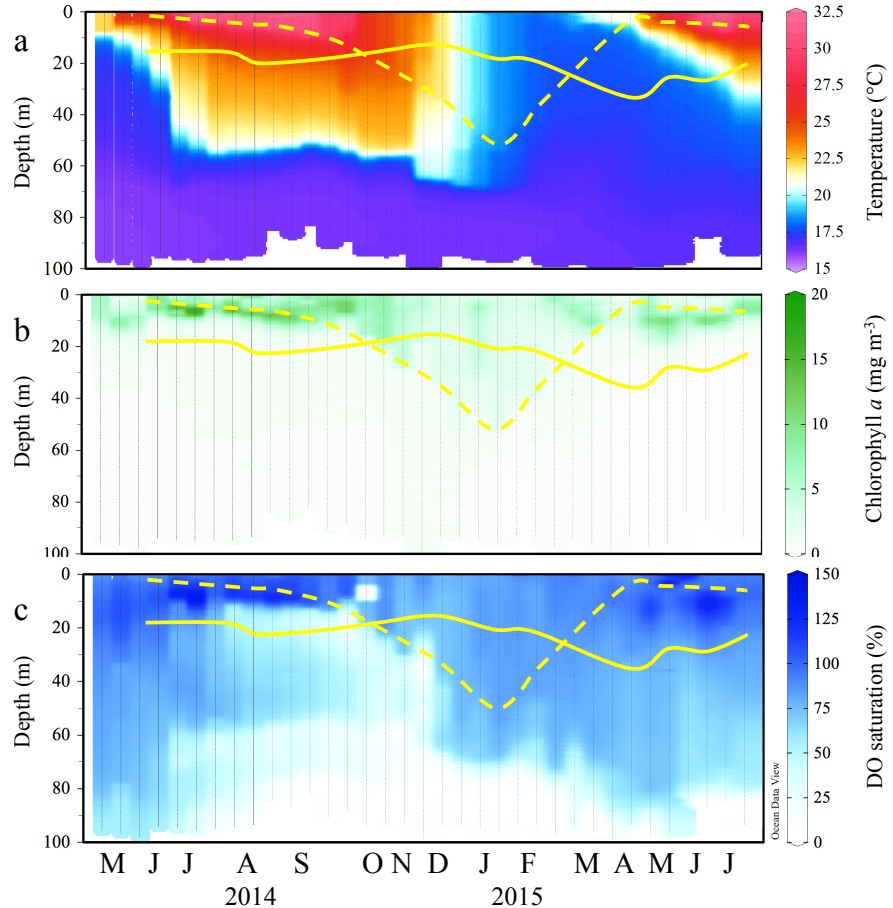

**Figure 2. Seasonal variability in a) temperature (°C), b) Chlorophyll a concentration (mg m$^{-3}$), and c) dissolved oxygen saturation (%) from S1 in the Feitsui Reservoir (S1). Profile data were normally collected on weekly basis throughout the warmer months and every two weeks in winter. Solid yellow line indicates the limit of euphotic zone and dashed yellow line the depth of mixed layer.**

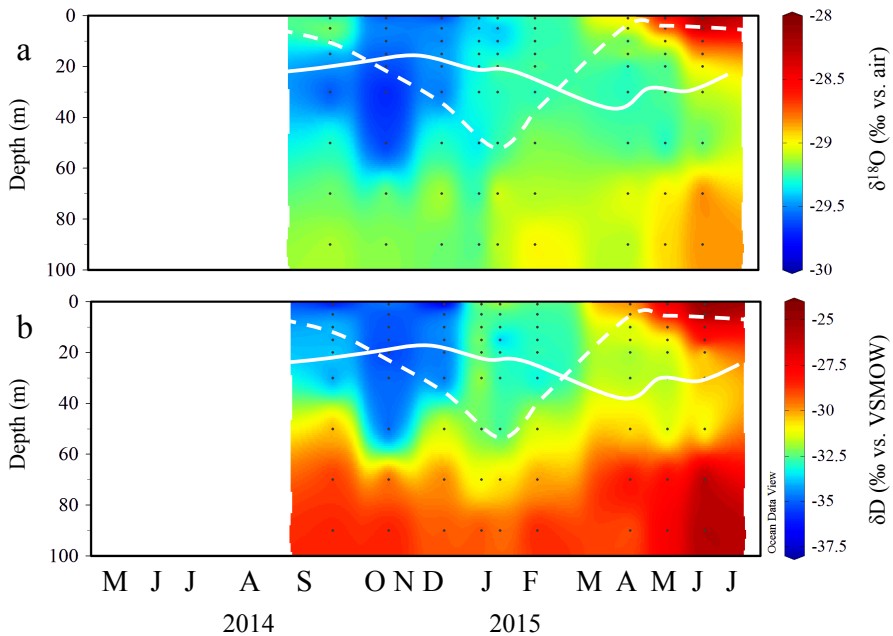

**Figure 3. Seasonal variability in a) δ$^{18}$O of water O$_2$ (‰, vs. VSMOW), and b) δD (‰, vs. VSMOW). Solid white line indicates the limit of euphotic zone and dashed white line the depth of mixed layer.**

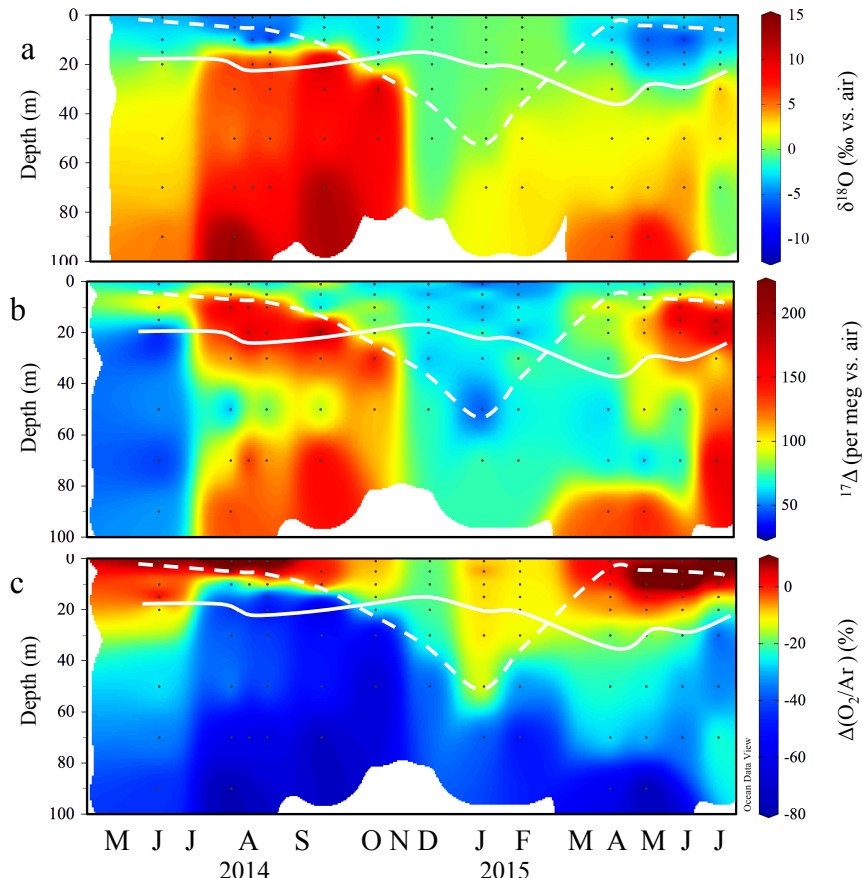

**Figure 4. Seasonal variability in a) δ¹⁸O of dissolved O₂ (‰, vs. air), b) ¹⁷Δ (per meg, vs. air) and c) Δ(O₂/Ar) (%) in Feitsui Reservoir. Solid white line indicates the limit of euphotic zone and dashed white line shows the depth of mixed layer.**

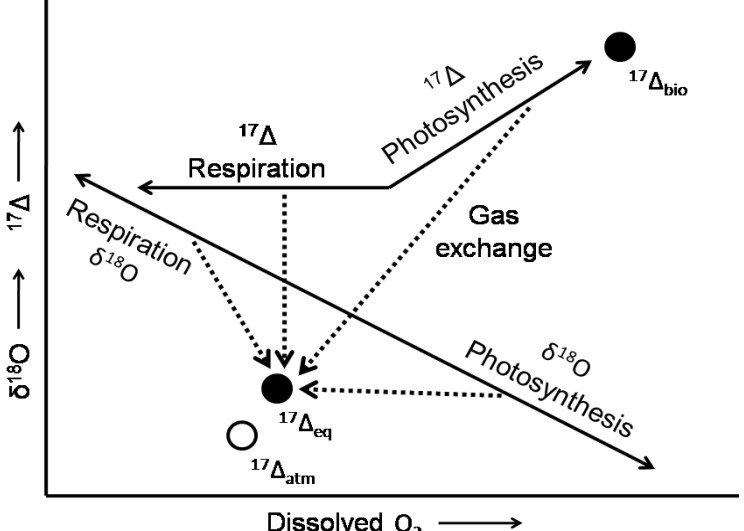

**Figure 5. A schematic diagram showing the effects of photosynthesis, respiration and air-water gas exchange on dissolved $O_2$ concentrations, $\delta^{18}O$ and $^{17}\Delta$. $\delta^{18}O$ changes with all the processes, including mixing. Because of non-mass dependent processes occurring in the stratosphere, the $^{17}\Delta$ of $O_2$ in air has a different signal to the $O_2$ produced biologically where fractionation is mass-dependent. $^{17}\Delta$ increases due to photosynthesis, decreases due to gas exchange but is not affected by respiration. Respiration removes $O_2$ and decreases the dissolved $O_2$ concentration but fractionates $O_2$ isotopes in a mass-dependent way, which does not affect the relative proportion of $\delta^{17}O$ and $\delta^{18}O$ and therefore the $^{17}\Delta$. $^{17}\Delta_{bio}$ is the maximum value of pure biological signal, which amounts to $^{17}\Delta$ of water. The slope of $^{17}\Delta$ increase towards $^{17}\Delta_{bio}$ is the kinetic slope $\lambda$ for respiration ($\lambda = 0.518$). $^{17}\Delta_{eq}$ is the $O_2$ at air-water equilibrium, which has a small offset from $^{17}\Delta_{atm}$, which is by definition 0, due to fractionation at equilibrium where $\delta^{17}O$ and $\delta^{18}O$ slopes during invasion and evasion follow a slightly different slope to that of respiration.**

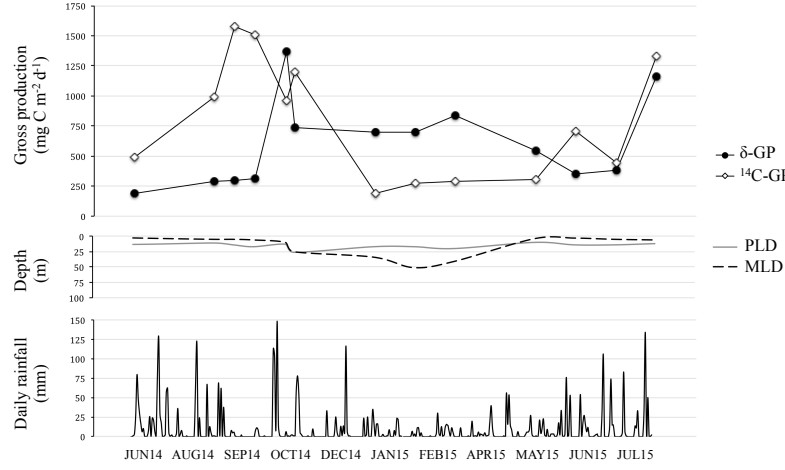

**Figure 6. Comparison between 'dual-delta' GP estimates (δ-GP) and $^{14}$C-GP rates. PLD and MLD indicate photosynthetic layer depth and mixed layer depth, respectively and precipitation shows the total daily rainfall.**

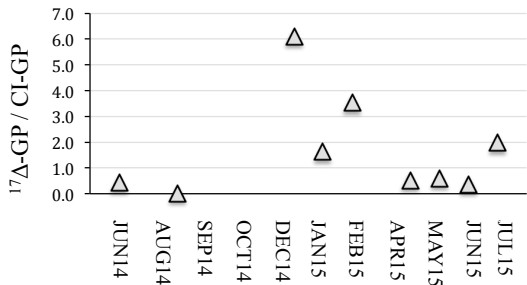

**Figure 7. The relative difference between production rates obtained from mixed layer model ($^{17}\Delta$-GP, following Luz and Barkan, 2000) and our whole column inventory approach (CI-GP).**

**Table 1. Air-equilibrated water results.**

| Deionised water sample | $\delta O_2/Ar$ (‰) vs. air | $\Delta(O_2/Ar)$ (%) | $\delta^{17}O$ (‰) vs. air | $\delta^{18}O$ (‰) vs. air | $^{17}\Delta^a$ (per meg) vs. air |
|---|---|---|---|---|---|
| 1 | -107 | -0.5 | 0.346 | 0.645 | 12 |
| 2 | -100 | 0.3 | 0.314 | 0.591 | 8 |
| 3 | -100 | 0.3 | 0.310 | 0.573 | 13 |

**Table 2. Summary of PP rates in the Feitsui Reservoir from June 2014 to July 2015. Dates marked with an asterisk indicate post-typhoon sampling days. $^{17}\Delta-GP$, $\delta-GP$, NP, $^{14}C-GP$ and CI−GP are in mg C m$^{-2}$ d$^{-1}$.**

| Date | Abbrev. | PLD[a] (m) | MLD[b] (m) | $C_o$ (mmol m$^{-3}$) | K from U (m day$^{-1}$) | $\Delta(O_2/Ar)$ (%) | $\Delta^{17}-GP$[c] | $\delta-GP$[d] | NP | NP/ $\delta-GP$ | $^{14}C-GP$ | CI-GP |
|---|---|---|---|---|---|---|---|---|---|---|---|---|
| 10/06/2014 | JUN14 | 14 | 3 | 249.01 | 0.21 | 10 | 120 | 187 | 44 | 0.24 | 492 | – |
| 19/08/2014 | AUG14 | 11 | 5 | 232.20 | 0.27 | 15 | 190 | 289 | 80 | 0.28 | 988 | 293 |
| 26/08/2014 | AUG14 | 14 | 5 | 232.20 | 0.30 | 13 | 193 | 297 | 78 | 0.26 | 1580 | 6314 |
| 02/09/2014 | SEP14 | 17 | 6 | 232.20 | 0.35 | 13 | 204 | 313 | 92 | 0.30 | 1510 | – |
| 23/09/2014* | SEP14 | 13 | 10 | 240.36 | 1.15 | 1 | 834 | 1372 | 28 | 0.02 | 961 | – |
| 28/10/2014 | OCT14 | 26 | 25 | 258.23 | 0.56 | -12 | 448 | 737 | -149 | -0.20 | 1197 | – |
| 09/12/2014 | DEC14 | 17 | 34 | 273.24 | 0.64 | -21 | 408 | 699 | -311 | -0.44 | 190 | – |
| 20/01/2015 | JAN15 | 17 | 51 | 289.89 | 0.65 | -9 | 397 | 696 | -140 | -0.20 | 275 | 67 |
| 10/02/2015 | FEB15 | 20 | 41 | 295.85 | 0.69 | -11 | 476 | 837 | -194 | -0.23 | 292 | 244 |
| 14/04/2015 | APR15 | 10 | 4 | 278.59 | 0.48 | 2 | 333 | 541 | 21 | 0.04 | 307 | 135 |
| 19/05/2015 | MAY15 | 14 | 3 | 244.62 | 0.40 | 6 | 223 | 351 | 49 | 0.14 | 708 | 628 |
| 23/06/2015 | JUN15 | 14 | 5 | 232.20 | 0.36 | 9 | 247 | 379 | 66 | 0.17 | 442 | 376 |
| 14/07/2015* | JUL15 | 12 | 6 | 240.36 | 1.00 | 11 | 758 | 1162 | 228 | 0.20 | 1328 | 657 |

[a] Photic layer depth. [b] Mixed layer depth. [c] Following Luz and Barkan (2000). [d] Following Prokopenko et al. (2011) and Kaiser (2011).