# Peer review of "Variations in triple isotope composition of dissolved oxygen and primary production in a subtropical reservoir"

_Biogeosciences, 2016_

## Referee Comment (RC1) · Anonymous Referee #1 · 29 Apr 2016

**Review for manuscript bg-2016-49:**

"Variations in triple isotope composition of dissolved oxygen and primary production in a subtropical reservoir" by:
Jurikova, H., T. Guha, O. Abe, F. Shiah, Ch. W and M. Liang

In this manuscript the authors present results of $O_2/Ar$ ratio measurements, as well as of stable oxygen isotopes in dissolved oxygen ($\delta^{17}O$ and $\delta^{18}O$) to estimate the $^{17}O$ excess ($^{17}\Delta$), and in water ($\delta^{D}$ and $\delta^{18}O$). Water samples were taken over a year (from May 2014 to July 2015), at different depths in the vertical water column of the Feitsui Reservoir, Taiwan. The authors used the oxygen measurements to estimate the net and gross production (NP and GP, respectively) to evaluate the reservoir's metabolic state and seasonal variability. This is the first time that the triple oxygen isotopes technique is applied in a freshwater enclosed system. The authors gathered a nice data set that can help to understand the fast changes of the metabolic balance in the reservoir, and prove the ability of the method to capture them. The manuscript is generally well written, however the structure has to be modified slightly, as well as the main focus of the paper. It lacks of strong linkages between the dominating physical factors in the reservoir (vertically and horizontally) and changes of $^{17}\Delta$ and $O_2/Ar$. Furthermore, it contains major flaws on the data processing of the data, such as corrections on the isotopic analysis of samples as well as in the calculation of GP from $^{17}\Delta$; this is in the first place not correctly done, and in the second place, I don't think this estimation can be applied to the reservoir due to its fast changing vertical water column dynamics which are not considered in the calculations. Therefore, at this point, I cannot accept this manuscript for publication. From my opinion, there are major changes that need to be done before this work can be considered for publication in Biogeosciences. Next, I list a summary of my major concerns:

1) The definition of mixed layer depth used by the authors is provided. This is a very relevant concept because it defines the physical limit for the NP and GP estimations based on $O_2/Ar$ and triple oxygen isotopes. Consideration of vertical transfer for GP and NP calculation cannot be neglected. Furthermore, in the way is given now, it makes totally irrelevant the calculation of GP since Feitsui Reservoir seems to have a complex vertical and horizontal water structure. The triple oxygen isotopes of dissolved oxygen, as well as the stable isotopes of water, can be powerful proxies that can be better used to understand the dynamics of the reservoir linked to their physical characteristics, and this is not sufficiently done in the manuscript in its current form.

2) Their statistical interpretation of the data lacks rigor and understanding of the method. The authors should express their precision and uncertainties in a better way. Also there is a lot of missing information in regard to the isotopic data correction due to interferences and imbalance between sample and reference side during the MS analysis, for example.

3) There seem to be a lack of explanation to the relevance of measuring aliquots of laboratory prepared equilibrated water. These serve as standard to estimate the reproducibility of the method during sample preparation and isotopic analysis by MS in the absence of samples duplicates, more detailed information is needed here.

4) After an improvement of the definition of mixed layer depth and interpretation of their changes, the GP calculation should be corrected to use $\delta^{18}O$ and $\delta^{17}O$ directly instead of $^{17}\Delta$. There is published evidence showing that due to numerical inaccuracies, this practice has to be changed.

5) Due to the fast changing physical dynamics in the reservoir, the authors should be careful in the GP estimation from triple oxygen isotopes, and this simply cannot be done in the same way as done until now for ocean basins.

Detailed information on these concerns, as well as a list of minor recommendations, are given below:

**Major comments:**
There is no place in the manuscript in which the authors mention their chosen criteria to define their mixed layer depth. It seems it follows nicely the Chl a vertical distribution in Fig. 2b, but this might be only and artifact of the colors in the figure. Whichever criteria the authors chose to define their mld seems just wrong and not necessarily useful for the observations in $^{17}\Delta$ values and estimation of mixed layer-GP. The timescale of processes that influence the vertical mixing in lakes and reservoirs depends on the basin size and stratification. I think here it is more complex to define a mixed layer depth that suits to the concept of the estimation of GP from oxygen isotopes. There seem to be a permanent and a temporal mixed layer, with overturning and convective cooling occurring at faster orders of magnitude than what can be estimated with a standard calculation for the gas exchange coefficient. The vertical displacement of primary producers and adaptation should be evaluated and taken into account together with the stable isotopes data. The mld definition for applications of GP from oxygen isotopes and NP from $O_2/Ar$ ratios must represent closely the metabolic state of the water column within the productive zone. A definition based on oxygen as done in Castro-Morales and Kaiser, 2012, could potentially help to define a better mld for GP and NP estimates based on oxygen measurements.
For this reason, I don't think here it is appropriate to apply the estimates of GP from the triple oxygen isotopes method for lakes and reservoirs as done for the ocean until now.

Besides: calculating GP from $^{17}\Delta$ should be avoided. This was the standard calculation and the approximation may be still fine for low, typically oceanic $^{17}\Delta$ values. However, higher values will lead to a larger error in the GP, to avoid this, GP should be instead calculated from the measured $\delta^{17}O$ and $\delta^{18}O$ as demonstrated by Prokopenko et al., 2011 and Kaiser, 2011. Since this are lake samples and some of the results show high $^{17}\Delta$ values, the authors should consider re-calculating their GP using directly the $\delta^{17}O$ and $\delta^{18}O$, but this should be done only if the authors find an agreement on defining a mixed layer adequate to the fast changes occurring in the water column of the reservoir. Furthermore, the presentation of $^{17}\Delta$ is a better practice in this case and should be presented and discussed in the manuscript as a proxy variable of GP.
The high oxygen supersaturation in the entire water column between May and June 2014 could be due to strong vertical mixing with most of the oxygen from

atmospheric source, this is also shown by the very low $^{17}\Delta$ and $\delta O_2/Ar$ values. Rain season?

Which physical processes (horizontal or vertical transfer) occur in the reservoir for the productivity to increase in July at depth? How relevant here are seiches? Was that specifically evaluated?

**P4:**

L10 – this part requires major explanation on the in vitro dissolved oxygen measurements, how many samples and at which depths were collected to calibrate the CTD data? What is the precision of the in vitro oxygen measurements? Which technique was used for the detection of the reduced oxygen species in the sample after titration?

L24 – This paragraph needs a reference for the extraction and collection method into the molecular sieve pellets. Did the authors follow Abe, 2009 (*Rapid Commun. Mass Spectrom.* 2008, 22, 2510) or Keedakkadan and Abe, 2015 (*Rapid Commun. Mass Spectrom.* 2015, 29, 775–781)?. If there was a modification to any of these two suggested extraction procedures, or it was used a different one (?) then the authors should specify in which way this was done.

**P5:**

L10-11 – this statistical analysis doesn't make sense. A student's t-test can be only applied for the comparison of normally distributed data sets. Here it is simply the average of the repetition of measurements (cycles) of the same sample (not duplicates or triplicates), which only gives the analytical precision or mean standard error in the measurement of one sample. It is clear that the more acquisitions with more cycles each will reduce the error in the measurement. The two-sigma outlier removal, why they were done? Were they the source of an error/contamination in the sample or in the IR measurement?. Statistically it doesn't mean anything to give an average of the standard error for all samples, since each sample is independent to each other in space and time, I encourage the authors to delete the sentence in L10-11.

Furthermore, the reproducibility and performance of the samples preparation and the MS analysis must be evaluated with the standard error of the air-equilibrated water samples mentioned only until the discussion section (P11, L23-25). This should be moved to section 2.3, see more comments below on this regard. Uncertainties in the $O_2/Ar$ ratio and $\delta^{18}O$ and $^{17}\Delta$ from the air-equilibrated water aliquots with respect to air should be also provided.

L20-21 – The explanation of this correction is not very clear. Did the authors corrected $\delta^{17}O$ due to $N_2$ interference in the analysis?

Also did the authors made corrections due to differential gas depletion between the sample and reference sides during the IR analysis? See Stanley et al., 2010 for this, and report if this was done or not.

A correction to $\delta^{18}O$ due to fractionation has to also be done, did the authors checked for this?

L17 – did the $\delta O_2/Ar$ were normalized to air? There is also no explanation regarding the correction for the residual gas in water sample after equilibration, the authors should have been done that in order to obtain their $([O_2]/[O_2]_{eq})_{bio}$ in eq. 3, but this is not stated in the manuscript. See also my comments below for P7 and Fig. 5.

**P8:**

L19 – I am missing more information from the results of the isotopic composition of water. I would have expected more insights on the different sources of water in the reservoir; it is possible to differentiate here rainwater and standing water? Is it possible to see here the extent of rainwater after typhoon events in the reservoir? How about possible horizontal contribution of water with potentially different characteristics?

L23 - Do the authors assume here that the $^{17}\Delta$ from JUL14 and AUG14 represent only biological production and these are selected to represent the end member $^{17}\Delta_{bio}$? The typical value used here was 249 per meg derived experimentally and shown in Luz and Barkan, 2000, and controversy on the values for the isotopic signatures of photosynthetic activity ($\delta^{17}O_p$ and $\delta^{18}O_p$) have been discussed in literature (e.g. Kaiser, 2011, Luz and Barkan, 2011 *Global Biogeochem. Cycles*, see also the comment by D.P. Nicholson (doi: 10.5194/bg-8-2993-2011) on the Kaiser (2011) paper for detailed discussion on this regard. The most recent values for $\delta^{17}O_p$ and $\delta^{18}O_p$ are shown in Luz and Barkan, 2011b (*Geophys. Res. Lett.*) and also in Barkan and Luz (2011, *Rapid Commun. Mass Spectrom.*) and in Kaiser and Abe (2012). The authors are encouraged to revise this literature and select a value for $\delta^{17}O_p$ and $\delta^{18}O_p$ in case they find a better definition of mixed layer depth for their GP estimates. In any case, the authors should stop using their $^{17}\Delta_{bio}$ for this.

**P9:**

L09 – By DEC14 thermal stratification nearly disappeared" but mld became deeper? How is mld defined? It follows the color code of Chl a in Fig. 2 but this might be an artifact of the figure. How vertical mixing is high but a very strong marked mld? What happened from SEP14 to OCT14 that the stratification broke, which physical process dominated for this change in the water column? (horizontal or vertical transfer? Why? Wind speed increase? Rain?), increase in atm $O_2$, but you cannot see that in $\delta O_2/Ar$, I recommend using $\Delta O_2/Ar$ instead.

L30 – is the nutrient availability also so high as in 100 m in spring 2015? Which physical process may dominate in the reservoir to achieve this?

L32 – How common are seiches in Feitsui Reservoir? This seems to be an important process that dominates the distribution of nutrients and gases vertically in the water column of the reservoir. More information is needed on this regard and the authors should discuss more this process in the context of their findings.

Which other external forcing processes the authors meant there?

**P10:**

L1 – were precipitation events recorded during the sampling periods (not only the typhoon events)?

L2-5 – here the authors suggest that vertical processes are relevant for the distribution of the $^{17}\Delta$ signal in the reservoir, it

L11-12 – most of the production seems to take place below the mld,

L12 – again this is arguable because of their definition of mld

**P11:**

L6 – how was that lifetime of $O_2$ estimated? Was based on mld and gas transfer coefficient?

L7 – I disagree that the vertical mixing and advection are negligible for $^{17}\Delta$ and ultimately GP determinations in the reservoir. I think the authors underestimate here the vertical transfer of biological $O_2$ (vertical displacement of primary producers within the reservoir) and their definition of mixed layer depth is by no means helpful for their budget model. The fact that there is high $^{17}\Delta$ in subsurface and deep waters defines the reservoir as a full column activity water system with marked seasonality and strong vertical influence, possibly due to wind and rain. It is hard to apply there the GP concept from $\delta^{17}O$ and $\delta^{18}O$ considering the shallow mixed layer depth. This is irrelevant to calculate here, I would be more focused on explaining the physical driving forces to the vertical distribution of $^{17}\Delta$, and how this changes in short periods of time the metabolic balance of the reservoir.

L8 – but most part of the sampling period the bottom limit of the euphotic zone lies below the mixed layer, so again the authors should revisit their definition of mld for GP calculations

L24-25 – more information is needed on the preparation of the air-equilibrated aliquots

**P12:**
L09-10 – I agree, but then why the authors wrote lines 10-12 in P. 11? This is contradictory to what is stated here. To actually compare productivity values based on $^{14}C$ and from oxygen isotopes, the sampling scheme had to be designed for this purpose with duplicate analysis and samples for $^{14}C$ also at depth.

**Minor comments:**
Throughout the manuscript, leave a space between the quantity and the unit in % and $^{o}/_{oo}$, and also for $^{o}C$

P1, L15 – add "water" reservoirs

P2, L11 – change "confining it to a small volume" to "confining them into a small volume"

P2, L16 – replace the symbol "&" by the word "and" here and all the citations throughout the manuscript where it is used

Modify to "introduced the triple oxygen-isotopes technique, …"

P2, L17 – change to "The $^{17}O$ excess is defined as:"

P2, L19 (eq. 1) – here and elsewhere all variables must be italicized, this is particularly the case of all delta symbols in: $^{17}\Delta$, $\delta^{17}O$ and $\delta^{18}O$, as well as $K$, $C_o$ introduced in eqs. 2 and 4, and throughout the manuscript.

P3, L3 – change "large" to "largely"

**P4:**
L1 – add comma between "quality" and "the watershed"

L1 – add "the" before Feitsui Reservoir

L2 – pluralize "area"

L2 – add "are" between "active" and "prohibited"

L3 – add "the" before Feitsui Reservoir

L4 – since when the meteorological station near Feitsui Reservoir has been active?

L5 – change from "processing" to "preparation"

L9 – change to "using a Sea-Bird CTD…", what was the vertical resolution of the CTD measurements?

L14 – leave a space between the number and the units (15 µL)

L19 – "…for removal of water vapor at liquid nitrogen temperature. The extracted gases…"

L21 – The GC is to separate $N_2$ and $CO_2$ from $O_2$ and Ar, please delete "contaminants". Please correct and complete the sentence by adding that only $O_2$ and Ar remain the main components in the gas mixture.

L22 – "During the separation … "

L25 – I suggest here to add a new section (2.3) that corresponds to the "Stable isotope analysis in water". Also an opening sentence to explain why this was done is needed, for example: "To identify the source of water in the reservoir, the $\delta^D$ and $\delta^{18}O$ in the $H_2O$ molecule of reservoir water was analyzed. For this, water samples were collected in 15 ml …."

L30- change uL to mL and to "an aliquot of 5 mL of water sample was converted to $O_2$ by injecting it to a $CoF_3$ reaction tube…". Leave also a space between 370 and $^oC$.

**P5:**

L3 – Do you mean that a set of duplicates of standard water samples were measured every 80 water samples analysis?

L8 – change to "$O_2$ from the purified oxygen-argon mixture (as explained in section 2.2)…"

L9 – change to "12 cycles each. Thus, the reported …"

L18 – this precision is for $\delta O_2/Ar$ in repetitions of atmospheric air measurements?

L23 – the correction is not to achieve high precision, it is simply a correction of the measurement due to interferences. Delete this sentence.

L26 – remove the second "of" (…"and for obtaining more precise results…")

L29 – How many samples represent one set or trip to the reservoir? Are all the black dots plotted in a single vertical profile in Fig. 5 representing one set of samples? They were not always the same number isn't?

**P6:**

L2-8 – what is written in this paragraph is only true for a system at steady state. This should be stated.

L16 – Co as expressed by the authors is not simply the $O_2$ solubility, but the $O_2$ concentration at saturation, or at equilibrium with the atmosphere, using the solubility coefficients from Benson and Krause, 1984, and the standard term to express this is $[O_2]_{sat}$.

L17 – did the daily wind speed measurements were collected from the meteorological station? Were they corrected to represent wind speed 10 m above sea level? Why averaged over 1 week? What is the residence time of the gas in the mixed layer depth of the reservoir as calculated from the gas transfer coefficient and the mixed layer depth?

L22-25 – this paragraph should be moved to another section maybe below section 2.2, stating specifically how the $^{14}C$ analysis was done.

L29 – change to "… Ar supersaturaion in water …"

**P7:**

L2, Eq. 3 –The term on the left hand side of Eq. 3 is misleading and doesn't represent what is really expressing. The biological O2 saturation should be expreseed as $\Delta O_2/Ar$ as in many past works that use this method (e.g. Cassar et al., 2011, Castro-Morales et al., 2013). Please avoid introducing new ways for terms and variables. The new

community using this method should make use of the same variables to express the terms to avoid confusion and to keep consistency.

L6-7 - here it should be stated that the authors corrected $\delta O_2/Ar$ for the residual gas in water sample after equilibration in order to obtain their $([O_2]/[O_2]_{eq})_{bio}$ (that should be $\Delta O_2/Ar$) in eq. 3.

L21 – I wouldn't call it permanent but seasonal stratification

L22 – the temperatures above 30 $^o$C are only in the top 10 m.

L25 – here it should be defined which criterion the authors used to define mixed layer depth.

Is only the change in atmospheric temperature what makes the temperature of the water reservoir to change? Is there no evidence of vertical or horizontal water transport? As mentioned later in the manuscript, other lake processes as the presence of seiches (P9, L32) or other external influences such as wind or water input from precipitation can also alter the temperature of the reservoir. Discuss this here in the context of this factors possibly contributing to the change in the water temperature. Of particular interest is what happens at depth in the reservoir, away of the direct atmospheric influence.

L26 – which processes occur within the reservoir to shallow the mixed layer depth in summer? Only warming by atmospheric influence at the surface?

L27 – in all other cases where mixed layer was present was also defined, only shallower

L28 – where are these sediments from? From the bottom or from lateral transport within the reservoir? Which other lake processes?

**P8:**
L13-16 – during late spring in 2014, $O_2$ supersaturation also at the bottom of the reservoir is seen, what is the origin of this? It should be then a very strong vertical mixing in the reservoir at this period of time, this is the indication of a vertical transfer also of potentially biological $O_2$. Why is so fast changing this to a very shallow $O_2$ supersaturation by end of April?

L22 – complete the sentence as: "showed more depleted values during autumn at the top 60 m…"

L23 – what do the authors mean with "selected waters"?

**P9:**
L5 – wouldn't be better JUL14 than JUN14?

L7 – Also in JUL14

L8-9 – the thermal stratification nearly disappeared, but the mixed layer depth just became deeper, how it is defined mld?

L22 –how much is the annual mean?

**P10:**
P10, L17-18 – remove one dot at the end of the sentence (it has two) and the dot at the end of line 18 should only have the comma.

L6-7 – change to "…we briefly discuss our results in the context of typhoon events."

L18 – higher wind speeds can also explain higher GP rates at depth of the reservoir?

**References**
- The reference of Barkan and Luz, 2005 is missing

**Figures**
**Fig. 2**, is one monthly band of data representing only a once in a month sampling? So this is not really an entire month of data but only few days (maybe only one day) in which sampling a set of samples took place? The interpolated figures as shown in Fig. 2, 3 and 5 are then very misleading, since the vertical data cannot be put sequentially one after the other and do a horizontal interpolation with them. There is a gap of about 29 days between them, and as it looks now in the figures, "fast" changes occur between one month and the other. I will be careful in the way the data is presented in this figures. I would rather do simply vertical profiles or not put together the bands. It is unclear to see how much of the information on the figure is the result of interpolation artifacts.
Also, the mixed layer depth and limit of euphotic zone should be drawn also from May 2014.

**Fig. 3**, what happened with the data from May to August 2014?
The lower $\delta^{18}O$ and $\delta^D$ from the surface to about 60 m from October to December 2014 is related to the first typhoon according to the authors, However, at the time of the second typhoon there is also a different d18O signal at the surface (top 20 m in May-July 2015), the authors must explain these differences and linkages to $\delta^{18}O$ and $\delta^D$ from dissolved oxygen as shown in Fig. 5a for the second typhoon.

**Fig. 5,**
**5b,** the low $^{17}\Delta$ in the water column from 20 m down during May-June 2014 is coincident with very high $O_2$ saturation, which indicates a strong vertical mixing from surface air saturated water down. This is also evidenced in the $\delta O_2/Ar$ signal. However, the authors claim that the high $^{17}\Delta$ seen at the depth during July-October 2014 also originates from vertical transfer from the surface, however, there is lower $^{17}\Delta$ signal in the top <10 m. could it be local $O_2$ photosyntetically produced? or horizontal transfer? Which process actually causes breaking down the high $^{17}\Delta$ in the entire column between 40 and 60 m from July-November 2014?
First signal of high $^{17}\Delta$ at the bottom (80-100 m) in March-May 2015 it seems is a different water mass, this is also seen in the $\delta^{18}O$ and $\delta O_2/Ar$, what is its origin? It looks lateral transport.

**5c**, why is this third depth point at around 20 m in August 2014 so high in $O_2/Ar$? Most of $\delta O_2/Ar$ is below zero. It is hard to see the biological and atmospheric contribution in this ratio. A better way to express this is as $\Delta O_2/Ar$ (biological $O_2$ saturation) in % (this is their $([O_2]/[O_2]_{eq})_{bio}$). I recommend the authors to plot instead $\Delta O_2/Ar$ in Fig. 5 panel c.

**Fig. 6**, are the $^{17}\Delta$ GP shown there is only the surface values?

---

## Referee Comment (RC2) · Anonymous Referee #2 · 11 May 2016

**bg-2016-49 review**

"Variations in triple isotope composition of dissolved oxygen and primary production in a subtropical reservoir" by:
Jurikova, H., T. Guha, O. Abe, F. Shiah, Ch. W and M. Liang

The authors report measurements of $O_2$/Ar, $^{17}O/^{16}O$ and $^{18}O/^{16}O$ ratios of dissolved gases in a fresh water reservoir. They sampled the water column for more than a year. In addition to dissolved gases, they also measured $^{18}O/^{16}O$, D/H and $^{17}O$ excess of water. Using these measurements they estimated gross and net primary production and their ratios (GP, NP and NP/GP respectively). In their estimates they applied the method introduced by Luz and Barkan in 2000 (LB00). LB00 demonstrated the potential of the method for both marine and fresh water studies. Since then the method has been used a number of times in marine systems but not in fresh water ones, so the data set collected is valuable in that it adds information on triple oxygen isotope variations in a freshwater system. This information has potential to help understanding the metabolic balance in lakes and fresh water reservoirs. Yet, there are a number of issues that need to be addressed before the material in the manuscript is suitable for publication.

In order to meaningfully interpret the results in quantitative terms of GP and NP, the authors need to realize that the LB00 method is applicable to mixed layer which is at steady state with respect to fluxes of photosynthesis, respiration and gas exchange with negligible effects of vertical and horizontal advection. While these conditions may be assumed for a number of marine situations, the reservoir in this manuscript may be more dynamic. If that's the case, to obtain meaningful quantitative estimates of GP and NP, the authors will need to include at least some of such dynamics in their calculations and apply a non-steady state model. While this may be a tall order, at the least, such approach should considered and discussed and the present estimates should be qualified and treated in a qualitative way. The data base of the study should be made available for future studies (see below) when a non-steady state model becomes available.

The authors are aware that a portion of the reservoir's photosynthesis takes place in the photic zone beneath the mixed layer. They have to give an estimate of how much is missing in their estimates for the mixed layer.

As well, in order to apply the LB00 method, it is necessary to know the $^{17}O$ excess of photosynthetic oxygen. While the latter depends on the $^{17}O$ excess of water, the two are not identical and the difference may be significant (see Luz and Barkan, 2011, GRL).

Even if the difference between $^{17}O$ excess of photosynthetic and water oxygens is known, I expect the value for water in the reservoir to be variable and to be dependent on fluctuations in the isotopic composition of meteoric water and evaporation from the reservoir. So more measurements of $^{17}O$ excess of water

are needed. The authors give one value for $^{17}O$ excess of water (246 per meg with respect to air). What are its $\delta^{17}O$ and $\delta^{18}O$ values?

Importantly, all raw data for $\delta^{17}O$ and $\delta^{18}O$ of dissolved and water oxygen should be given in tables suitable for web appendix if the paper is published.

---

## Author Comment (AC1) · 25 Jun 2016

Dear Editor and Reviewers,

On behalf of all co-authors, thank you very much for the comments and for handling our manuscript. The reviewers have spent considerable amount of time on this review and we very much appreciate their efforts. In reviewing all the comments and concerns raised, there are four major criticisms: mixed layer definition (mixed layer approach to approximate gross productivity), steady state assumption, lacking interpretation on the dynamics of the reservoir and insufficient technical details or data processing. Our responses for the former two are briefly summarized below. For the later two points, our response may be briefly summarized as follows; below we presented additional details to clarify reviewers' queries, we recalculated the GP estimates, where applicable we highlighted our previous peer-reviewed publications for further details, we acquired new data (i.e. meteorological), and are presently analysing more samples (i.e. water samples). All other details critical to the work will be included in the revised manuscript.

Mixed layer and mixing in the reservoir: Feitsui reservoir is a typical monomictic system that stays thermally stratified throughout the greater part of the year. The topographic characteristics of the reservoir, a large water mass located in a valley make its physical structure fairly simple and stable over the seasonal scale. The water residence time in the reservoir is rather long; throughout our study we estimated it to be about 150 days, sufficiently long to mix horizontally well. Previously reported field measurements as well as a model simulation showed alike trends in dissolved oxygen concentration throughout the period of 12 months, recorded at our sampling location and at Wu-Tan station situated on the other side of the reservoir upstream of the Beishi Creek in the top, the middle and the bottom layer of the water column, supporting the argument that our sampling station is horizontally rather uniform and not affected significantly by horizontal water advection. Regarding the mixed layer definition, the mixed layer was determined based on visual inspections of vertical temperature profiles. We opted for this method to ensure that the well mixed epilimnion only, homogenous in temperature and dissolved oxygen concentration is considered as mixed layer, without any influences from the thermocline. Alternative definitions were also explored and examined and are presented below. See also the response to the steady state approximation below, in support of the mixed layer approximation.

Steady state approximation: We showed below that steady state model is a good approximation for the freshwater Feitsui reservoir, by evaluating the total inventory (the column inventory approach shown below) from the whole column calculations.

We have also made substantial modifications that to the manuscript following reviewers' comments and suggestions and added a supplement with all sample data used in this study that will be ready for submission, providing an encouraging decision is made.

Below, please find our point-by-point response (in blue) to referee's comments (referee's comments are in italics).

Sincerely yours,

Mao-Chang Liang
Academia Sinica

**Response to reviewers' comments on manuscript bg-2016-49 ("Variations in triple isotope composition of dissolved oxygen and primary production in a subtropical reservoir") by Jurikova H., T. Guha, O. Abe, F.-K. Shiah, C.-H. Wang and M.-C. Liang**

**Reply to comments by reviewer #1**

*In this manuscript the authors present results of O2/Ar ratio measurements, as well as of stable oxygen isotopes in dissolved oxygen (δ17O and δ18O) to estimate the 17O excess (17Δ), and in water (δD and δ18O). Water samples were taken over a year (from May 2014 to July 2015), at different depths in the vertical water column of the Feitsui Reservoir, Taiwan. The authors used the oxygen measurements to estimate the net and gross production (NP and GP, respectively) to evaluate the reservoir's metabolic state and seasonal variability. This is the first time that the triple oxygen isotopes technique is applied in a freshwater enclosed system. The authors gathered a nice data set that can help to understand the fast changes of the metabolic balance in the reservoir, and prove the ability of the method to capture them. The manuscript is generally well written, however the structure has to be modified slightly, as well as the main focus of the paper. It lacks of strong linkages between the dominating physical factors in the reservoir (vertically and horizontally) and changes of 17Δ and O2/Ar.*
*Furthermore, it contains major flaws on the data processing of the data, such as corrections on the isotopic analysis of samples as well as in the calculation of GP from 17Δ; this is in the first place not correctly done, and in the second place, I don't think this estimation can be applied to the reservoir due to its fast changing vertical water column dynamics which are not considered in the calculations.*
*Therefore, at this point, I cannot accept this manuscript for publication. From my opinion, there are major changes that need to be done before this work can be considered for publication in Biogeosciences. Next, I list a summary of my major concerns:*

*1) The definition of mixed layer depth used by the authors is provided. This is a very relevant concept because it defines the physical limit for the NP and GP estimations based on O2/Ar and triple oxygen isotopes. Consideration of vertical transfer for GP and NP calculation cannot be neglected. Furthermore, in the way is given now, it makes totally irrelevant the calculation of GP since Feitsui Reservoir seems to have a complex vertical and horizontal water structure.*

**Reply #1**
Feitsui reservoir is a typical monomictic system (characteristic of subtropical lakes), that stays thermally stratified throughout the greater part of the year, with changing intensity of winter vertical mixing depending on the meteorological conditions. The topographic characteristics of the reservoir, a large water mass located in a valley make its physical structure (i.e. water temperature) fairly simple and stable over the seasonal scale (Itoh et al. 2015, their Fig. 2d). The water residence time in the reservoir is rather long, throughout our study we estimated it to be about 150 days, comparable to durations reported in the past (150 days reported by Kuo et al. 2003 and 115 days reported by Chen et al. 2006), sufficiently long to mix horizontally well. Field measurements as well as model simulation by Kuo et al. 2003 report alike trends in dissolved oxygen concentration throughout the period of 12 months, recorded at the

Dam site (S1 – the sampling station used in our study) and at Wu-Tan station located on the other side of the reservoir upstream of the Beishi Creek in the top, the middle and the bottom layer of the water column (Kuo et al. 2003 Fig. 6e and 7e), supporting the argument that our sampling station is horizontally rather uniform and not affected significantly by horizontal water advection. For further details on the mixed layer please see reply #3 below.

To assess the relevance of mixed layer model for estimating GP by isotope mass balance calculation, we test an alternative mass balance model based on the column inventory approach (onwards referred to as column inventory approach). Unlike isotope mass balance limited to the the mixed layer, the column inventory model requires time-series data of full profile from the surface to the bottom of the lake, but is able to obtain GP below mixed layer, which occurs when depth of photic zone is below the bottom of mixed layer, without steady-state assumptions.

Calculating the GP by the column inventory model is done by solving the following simultaneous equations:

$$^{16}O_t - {}^{16}O_{t-1} = {}^{16}P - {}^{16}C + {}^{16}I - {}^{16}E \text{ (eq. 1)}$$
$$^{17}O_t - {}^{17}O_{t-1} = {}^{17}P - {}^{17}C + {}^{17}I - {}^{17}E \text{ (eq. 2)}$$
$$^{18}O_t - {}^{18}O_{t-1} = {}^{18}P - {}^{18}C + {}^{18}I - {}^{18}E \text{ (eq. 3)}$$

where $^{n}O_t$ and $^{n}O_{t-1}$ are total amount of oxygen isotope n in the water column from surface to bottom of lake at time slice t and t-1 (just before time t), respectively; $^{n}P$, $^{n}C$, $^{n}I$ and $^{n}E$ are GP, consumption rate for entire water column, influx from the atmosphere and efflux to the atmosphere, respectively for oxygen isotope n. Eq. 1 can be substituted by column inventory or rates of total dissolved oxygen (eq. 4).

$$O_t - O_{t-1} = P - C + I - E \text{ (eq. 4)}$$

Eqs. 2 and 3 can be obtained by multiplying isotopic composition $(1 + \delta^{n}O)$ and/or isotope fractionation factor $(1 + {}^{n}\varepsilon)$ on eq. 4.

Fig. 1 shows the results obtained by the column inventory method as well as by mixed layer methods. By comparing blue lines with red diamonds, these are in agreement each other, except for late spring in 2015. This may indicate that the mixed layer model could be valid in the Feitsui Reservoir, not only in the open ocean stations.

[Figure]

**Fig. 1. Summary of GP results obtained from three different stable isotope mass balance calculations. Grey diamonds represent the results obtained by $^{17}\Delta$ mass balance model proposed by Luz and Barkan (2000). Black diamonds show results using the same mass balance model, but changing the $^{17}\Delta_{bio}$ from 246 per meg (corresponding to Feitsui reservoir water $^{17}\Delta$) to $^{17}\Delta_{bio}$ to 272 per meg (accounting for the difference between $^{17}\Delta$ of water and $^{17}\Delta$ of phytoplankton, for further details please see our reply #16 below). Red triangles represent the results obtained by mass balance model using dual-δ methods later proposed by Prokopenko et al. (2011) and Kaiser (2011), with $\delta^{17}O_{P-FR}$ and $\delta^{18}O_{P-FR}$ values as described in reply #16 below. The difference in the GP rates between the black diamonds and the red triangles therefore illustrate the offset between Luz and Barkan (2000) model and the dual-δ method. Blue lines are results obtained by column inventory method. Unlike other methods, values by column inventory calculation are averaged temporarily, so that they are not indicated by dots but by horizontal lines. Values from September to December in 2014 are not available due to unavailability of dissolved oxygen profile.**

Based on this we have made amendments to the manuscript; we have included the column inventory approach interpretation and compared the results between the different methods. A more thorough description on the mixing in the Feitsui reservoir is also made.

Relevant references:
Itoh M, Kobayashi Y, Chen TY, Tokida T, Fukui M, Kojima H, Miki T, Tayasu I, Shiah FK, and Okuda N. 2015. Effect of inter-annual variation in winter vertical mixing on CH4 dynamics in a subtropical reservoir. J. Geophys. Res. – Biogeosciences. 120 (7): 1177–1195.

Kuo, J.-T., Wen-Cheng Liu, Ruey-Tyng Lin, Wu-Seng Lung, Ming-Der Yang, Chou-Ping Yang, and Show-Chyuan Chu, 2003. Water quality modeling for the Feitsui

Reservoir in northern Taiwan. Journal of the American Water Resource Association 39(4): 671-687.

*The triple oxygen isotopes of dissolved oxygen, as well as the stable isotopes of water, can be powerful proxies that can be better used to understand the dynamics of the reservoir linked to their physical characteristics, and this is not sufficiently done in the manuscript in its current form.*

**Reply #2**
In the amended manuscript, we have expanded the discussion on the linkages between observed variation in the $^{17}\Delta$ and $\Delta O_2/Ar$ as well as the isotopic composition of water in the reservoir and physical processes including changes in thermal stratification, wind stress and precipitation, thanked to the newly requested additional data from the Feitsui Reservoir Administration.

*2) Their statistical interpretation of the data lacks rigor and understanding of the method. The authors should express their precision and uncertainties in a better way. Also there is a lot of missing information in regard to the isotopic data correction due to interferences and imbalance between sample and reference side during the MS analysis, for example.*

**Reply:** Please see below, this is addressed in multiple responses where specific issues where highlighted.

*3) There seem to be a lack of explanation to the relevance of measuring aliquots of laboratory prepared equilibrated water. These serve as standard to estimate the reproducibility of the method during sample preparation and isotopic analysis by MS in the absence of samples duplicates, more detailed information is needed here.*

**Reply:** Please see below reply #10.

*4) After an improvement of the definition of mixed layer depth and interpretation of their changes, the GP calculation should be corrected to use $\delta^{18}O$ and $\delta^{17}O$ directly instead of $^{17}\Delta$. There is published evidence showing that due to numerical inaccuracies, this practice has to be changed.*

**Reply:** Please see below reply #4.

*5) Due to the fast changing physical dynamics in the reservoir, the authors should be careful in the GP estimation from triple oxygen isotopes, and this simply cannot be done in the same way as done until now for ocean basins.*

**Reply:** Please see reply #1 as well as reply #4.

*Detailed information on these concerns, as well as a list of minor recommendations, are given below:*

***Major comments:***

*There is no place in the manuscript in which the authors mention their chosen criteria*

*to define their mixed layer depth. It seems it follows nicely the Chl a vertical distribution in Fig. 2b, but this might be only and artifact of the colors in the figure. Whichever criteria the authors chose to define their mld seems just wrong and not necessarily useful for the observations in $^{17}\Delta$ values and estimation of mixed layer-GP. The timescale of processes that influence the vertical mixing in lakes and reservoirs depends on the basin size and stratification. I think here it is more complex to define a mixed layer depth that suits to the concept of the estimation of GP from oxygen isotopes. There seem to be a permanent and a temporal mixed layer, with overturning and convective cooling occurring at faster orders of magnitude than what can be estimated with a standard calculation for the gas exchange coefficient. The vertical displacement of primary producers and adaptation should be evaluated and taken into account together with the stable isotopes data. The mld definition for applications of GP from oxygen isotopes and NP from $O_2$/Ar ratios must represent closely the metabolic state of the water column within the productive zone. A definition based on oxygen as done in Castro-Morales and Kaiser, 2012, could potentially help to define a better mld for GP and NP estimates based on oxygen measurements.*

*For this reason, I don't think here it is appropriate to apply the estimates of GP from the triple oxygen isotopes method for lakes and reservoirs as done for the ocean until now.*

**Reply #3**

The mixed layer was determined based on visual inspections of vertical temperature profiles. We opted for this method to ensure the homogenous epilimnion only is considered as the mixed layer without influences from the thermocline.

We appreciate the suggestions for an improved definition of the mixed layer. In the figure below we compare our mixed layer definition (hereafter MLD-T) and the "1% difference with respect to near surface (~3 bar)" (MLD-DO) from Castro-Morales and Kaiser (2012). Although Castro-Morales and Kaiser (2012) conclude the "0.5% difference with respect to 10 dbar" definition is a more optimal choice in their study, it may not be suitable for subtropical lakes, where the summer stratification often takes place above the reference depth of 10 dbar and therefore could result in overestimating the mixed layer depth. Both, the MLD-T and the MLD-DO show a very close agreement for the summer months. From October to February the MLD-DO tends to underestimate the mixed layer, likely as a result of oxygen variations that are slightly higher than 1% with respect to the reference depth 3 dbar. Nonetheless, as we show in the figure below with vertical profiles of temperature and dissolved oxygen concentration, the same mixed layer depth result may be obtained from visual assessment of the temperature profiles as well as the dissolved oxygen concentration profiles. The thermal stratification of the reservoir must therefore be representative of any changes in dissolved $O_2$ in the water column. We have now clarified the mixed layer definition in the manuscript.

[Figure]

[Figure]

*Besides: calculating GP from $^{17}\Delta$ should be avoided. This was the standard calculation and the approximation may be still fine for low, typically oceanic $^{17}\Delta$ values. However, higher values will lead to a larger error in the GP, to avoid this, GP should be instead calculated from the measured $\delta^{17}O$ and $\delta^{18}O$ as demonstrated by Prokopenko et al., 2011 and Kaiser, 2011. Since this are lake samples and some of the results show high $^{17}\Delta$ values, the authors should consider re-calculating their GP using directly the $\delta^{17}O$ and $\delta^{18}O$, but this should be done only if the authors find an agreement on defining a mixed layer adequate to the fast changes occurring in the water column of the reservoir. Furthermore, the presentation of $^{17}\Delta$ is a better practice in this case and should be presented and discussed in the manuscript as a proxy variable of GP.*

**Reply #4**

We have recalculated the GP based on the $\delta^{17}O$ and $\delta^{18}O$ values of dissolved oxygen measurements following the dual delta method by Prokopenko et al. 2011 and Kaiser 2011. For comparison, we also included the GP rates from $^{17}\Delta$ (the Luz and Barkan 2000 approach). We now observe a significant decrease in the GP rates, in particular during the winter months (see the Fig 1. in reply #1) comparing to the GP estimates

from $^{17}\Delta$. Moreover, the dual delta GP rates are in agreement with the results from the column inventory approach. These results, not only show the extent of the error due to the use of the $^{17}\Delta$ approximation instead of the dual delta method, but also confirm the robustness of the mixed layer method for Feitsui reservoir. We have amended this in the revised manuscript; we now use the dual delta method for calculation of GP rates, but also present a comparison between the approaches. Although no longer used for the calculation of GP rates, we expanded the discussion on the $^{17}\Delta$ as a valuable proxy and tracer of photosynthetic $O_2$ alone.

*The high oxygen supersaturation in the entire water column between May and June 2014 could be due to strong vertical mixing with most of the oxygen from atmospheric source, this is also shown by the very low $^{17}\Delta$ and $\delta O_2/Ar$ values. Rain season?*

**Reply #5**
We thank the reviewer for pointing out this. After examining the high supersaturation months, we found inconsistency between the reported $O_2$ content from CTD and that from $\delta O_2/Ar$. We then inspected the CTD data and found that the calibration was not done properly. We apologize for the mistake and thank the reviewer for indicating this point. The updated figure with dissolved $O_2$ saturation is shown below and is used to replace the wrong one in the revised manuscript.

Nonetheless, in May and June 2014 we still observe high $O_2$ saturation throughout the whole water column. To properly answer this question, we have analyzed the precipitation record from the Feitsui reservoir meteorological station. In Taiwan, the raining season, Meiyu, starts in May. As a result vertical mixing is more efficient, affecting the level of photosynthetic $O_2$ accumulating in water column. Furthermore, year 2014 was much warmer than year 2015, affecting both, the thermal stratification as well as the precipitation and making year 2014 distinctive from year 2015.

[Figure]

*Which physical processes (horizontal or vertical transfer) occur in the reservoir for the productivity to increase in July at depth? How relevant here are seiches? Was that specifically evaluated?*

**Reply #6**

Seiches were never evaluated in the Feitsui reservoir and to investigate them is beyond the scope of our present study; however, we would like to stress the importance of considering these phenomena in future studies to understand their role in distribution of gases and nutrients in the water column in the Feitsui reservoir. Alternatively, intrusion of surface water to about 50 m depth can also be supported by dust loading, where we see an increased amount of total suspended material (TSM) concentration (please see in the figure below). This intrusion however occurs at upstream tributaries at distance far away from the reservoir, resulting in rather uniform horizontal profiles across the reservoir reported earlier, for example, by Kuo et al. (2003).

The physical forces that cause vertical mixing in the reservoir include instabilities caused by heat losses at the surface, wind stress at the air-water interface, and instabilities caused by shear at the thermocline. In JUL15 a typhoon Chan Hom closely affected the reservoir area, likely enhancing the vertical transfer (seen from dissolved $O_2$ saturation and $\delta O_2/Ar$) and elevating primary production.

[Figure]

*P4:*
*L10 – this part requires major explanation on the in vitro dissolved oxygen measurements, how many samples and at which depths were collected to calibrate the CTD data? What is the precision of the in vitro oxygen measurements? Which technique was used for the detection of the reduced oxygen species in the sample after titration?*

**Reply #7**
Water samples collected from 10 depths (0, 2, 5, 10, 15, 20, 30, 50, 70 and 90 m) via 5-L GO-FLO samplers were siphoned into triplicate 60 ml bottles for dissolved oxygen analysis (Wheaton). A colorimetric method of Pai et al. (1993) was adopted for in vitro dissolved $O_2$ measurements with precision of 0.2 % r.s.d. (full scale).

We have added this to the manuscript and it now reads "Standard vertical profiles of conductivity, temperature and pressure were obtained routinely using Ocean Seven 316 CTD (IDRONAUT, Italy) with sensors for fluorescence and dissolved oxygen.

The accuracy of dissolved oxygen measurements was verified against in vitro measurements. For this, samples were siphoned intro triplicate 60 ml bottles (Wheaton) and a colorimetric method of Pai et al. (1993) was adopted for in vitro dissolved $O_2$ determination with precision of 0.2 % r.s.d. (full scale)."

Relevant references:
Pai SC, Gong GC and Liu KK. 1993. Determination of dissolved oxygen in seawater by direct spectrophotometry of total iodine. Marine Chemistry (41):343-351.

*L24 – This paragraph needs a reference for the extraction and collection method into the molecular sieve pellets. Did the authors follow Abe, 2009 (Rapid Commun. Mass Spectrom. 2008, 22, 2510) or Keedakkadan and Abe, 2015 (Rapid Commun. Mass Spectrom. 2015, 29, 775–781)? If there was a modification to any of these two suggested extraction procedures, or it was used a different one (?) then the authors should specify in which way this was done.*

**Reply #8**
We followed Abe (2009) with slight modifications. The O2-Ar mixture was absorbed on 2 pellets of molecular sieve (1.6 mm, 5A, manufactured by SUPELCO). It now reads "The separation was done using a chromatographic column (3 m long, 1/8" SS tube, with molecular sieve 5A at mesh 60/100), modified from Barkan & Luz (2003). For sample extraction and collection we have followed Abe (2009) with slight modifications. During separation the chromatographic column was kept at room temperature, and the yielded oxygen-argon mixture was absorbed onto two pellets of molecular sieve (1.6 mm, 5A, manufactured by SUPELCO) for subsequent isotopic analysis."

*P5:*
*L10-11 – this statistical analysis doesn't make sense. A student's t-test can be only applied for the comparison of normally distributed data sets. Here it is simply the average of the repetition of measurements (cycles) of the same sample (not duplicates or triplicates), which only gives the analytical precision or mean standard error in the measurement of one sample. It is clear that the more acquisitions with more cycles each will reduce the error in the measurement. The two-sigma outlier removal, why they were done? Were they the source of an error/contamination in the sample or in the IR measurement?. Statistically it doesn't mean anything to give an average of the standard error for all samples, since each sample is independent to each other in space and time, I encourage the authors to delete the sentence in L10-11.*

**Reply #9**
We first thank the reviewer for pointing this out. We reported the results following Luz and Barkan (2003). In the revised manuscript, we describe more clearly how the error is defined.

*Furthermore, the reproducibility and performance of the samples preparation and the MS analysis must be evaluated with the standard error of the air-equilibrated water samples mentioned only until the discussion section (P11, L23-25). This should be moved to section 2.3, see more comments below on this regard. Uncertainties in the O2/Ar ratio and δ18O and 17Δ from the air-equilibrated water aliquots with respect to air should be also provided.*

**Reply #10**
Here, we would like to point to an overseen error in the original manuscript; the air-equilibrated water was prepared using a "stirring method" not a "bubbling method" as stated. The equilibration was achieved by stirring of 8 L of deionised water with added $HgCl_2$ in a circulator with temperature control at 25 °C over the period of 72 hours. Dissolved gases were extracted following the same procedure as applied for the reservoir sample collection. The precision for analyses of individual equilibrated water samples was 0.020 ‰, 0.037 ‰ and 3 per meg for $\delta^{17}O$, $\delta^{18}O$, and $^{17}\Delta$, respectively and better than 4 ‰ for $\delta O_2/Ar$. For comparison, the long-term precision for routine measurements of atmospheric air was 0.017 ‰, 0.030 ‰, and 6 per meg, for $\delta^{17}O$, $\delta^{18}O$, and $^{17}\Delta$, respectively and better than 5 ‰ for $\delta O_2/Ar$. We have moved this to section 2.3 now, and expanded on the description of preparation of air-equilibrated water samples as well as on the uncertainties in the $O_2/Ar$ ration, $\delta^{17}O$, $\delta^{18}O$ and $^{17}\Delta$.

*L20-21 – The explanation of this correction is not very clear. Did the authors corrected δ17O due to N2 interference in the analysis?*

**Reply #11**
We checked regularly the signals at m/z '40' and m/z '28'. This is to verify the purity of the collected oxygen-argon mixture after the GC separation and/or to exclude any potential leak of atmospheric air during the introduction of the sample to the inlet of the mass spectrometer. We did not detect any significant presence of $N_2$, during the measurement of our samples, either dissolved oxygen or atmospheric air samples. For all samples the $N_2/O_2$ signal ratio was lower than 0.001. The same ratio was also found in atmospheric air samples used for used for referencing the reported dissolved $O_2$ isotopic composition and content. Even in case of any minor influences from the presence of $N_2$, this effect would accounted for when reporting the data in reference to air $O_2$. Therefore, no correction due to $N_2$ was applied. We have now clarified this in the manuscript.

*Also did the authors made corrections due to differential gas depletion between the sample and reference sides during the IR analysis? See Stanley et al., 2010 for this, and report if this was done or not. A correction to δ18O due to fractionation has to also be done, did the authors checked for this?*

**Reply #12**
We did not perform any corrections due to differential gas depletion between the bellows, as we also did not observe any fractionation (within the current precision). We performed routine analysis of atmospheric air (before and after sample analyses), which results are also included in the supplement.

*L17 – did the δO2/Ar were normalized to air? There is also no explanation regarding the correction for the residual gas in water sample after equilibration, the authors should have been done that in order to obtain their ([O2]/[O2]$_{eq})_{bio}$ in eq. 3, but this is not stated in the manuscript. See also my comments below for P7 and Fig. 5.*

**Reply #13**
Yes, we referenced the $\delta O_2/Ar$ to air. Given that the corrections (~2 ‰ and 0.02 ‰,

for $\delta O_2/Ar$ and $\delta^{18}O$, respectively) are less than the current analytical precisions (5 ‰ and 0.03 ‰, for $\delta O_2/Ar$ and $\delta^{18}O$, respectively), we did not apply the corrections. In the revised manuscript we will apply the corrections following Luz et al. (2002).

Relevant references:
Luz et al. (2002, Limnology and Oceanography 47, 33-42)

*P8:*

*L19 – I am missing more information from the results of the isotopic composition of water. I would have expected more insights on the different sources of water in the reservoir; it is possible to differentiate here rainwater and standing water? Is it possible to see here the extent of rainwater after typhoon events in the reservoir? How about possible horizontal contribution of water with potentially different characteristics?*

**Reply #14**
Please see further details in reply #1 and #6. It is unlikely for the variations to be a result of horizontal contribution of water with different isotopic composition (please see response #1). The explicit water source is largely standing water, given the rather long water residence time (~150 days) as compared to the mixed layer mixing time (~6 to 54 days). We have expanded the discussion on this in the revised manuscript.

*L23 - Do the authors assume here that the 17Δ from JUL14 and AUG14 represent only biological production and these are selected to represent the end member 17Δbio?*

**Reply #15**
Due to the small variations in $\delta^{18}O$ and $\delta D$, and long residence time (~150 days) of the reservoir water, we do not expect to find high variations in $^{17}\Delta$ of the water. For comparison, summer time values (JUL14 and AUG14) of $^{17}\Delta$ of the water in the reservoir were 38 ± 9 per meg (at lambda slope 0.528, vs. VSMOW), while September 2015 and July 2015 the $^{17}\Delta$ values of tap water were 38 ± 9 per meg and 30 ± 7 per meg, respectively. However, to properly address reviewers' concern on this we are currently analysing archived water samples that will enable us to fully constrain any potential seasonal variations in $^{17}\Delta$ of the water in the reservoir.

*The typical value used here was 249 per meg derived experimentally and shown in Luz and Barkan, 2000, and controversy on the values for the isotopic signatures of photosynthetic activity (δ17Op and δ18Op) have been discussed in literature (e.g. Kaiser, 2011, Luz and Barkan, 2011 Global Biogeochem. Cycles, see also the comment by D.P. Nicholson (doi: 10.5194/bg-8-2993-2011) on the Kaiser (2011) paper for detailed discussion on this regard. The most recent values for δ17Op and δ18Op are shown in Luz and Barkan, 2011b (Geophys. Res. Lett.) and also in Barkan and Luz (2011, Rapid Commun. Mass Spectrom.) and in Kaiser and Abe (2012). The authors are encouraged to revise this literature and select a value for δ17Op and δ18Op in case they find a better definition of mixed layer depth for their GP estimates. In any case, the authors should stop using their 17Δbio for this.*

**Reply #16**

For the $^{17}\Delta_{bio}$ we used a value 246 per meg, which we determined based on measurements of water samples collected in the reservoir in July and August 2014. Isotopic variations of seawater from different parts of the ocean are small and therefore an average $^{17}\Delta_{bio}$ 249 ± 15 per meg may be representative for the ocean. This is however not the case for freshwater systems, where the isotopic composition of water tends to vary geographically and among different water sources (Luz and Barkan 2010, GCA). Thus, $^{17}\Delta_{bio}$ or $\delta^{17}O_P$ and $\delta^{18}O_P$ may not be taken from the literature for the Feitsui Reservoir and should be determined in situ. For comparison, Luz and Barkan 2000 give $^{17}\Delta_{bio}$ 159 per meg for the freshwater Lake Kinneret, the only up-to-date $^{17}\Delta_{bio}$ for a lake in the literature.

However, as pointed out by the reviewer #2 also, the $^{17}O$-excess of water is not identical to the $^{17}O$-excess of photosynthetic $O_2$, which we previously did not account for. In the revised manuscript, we follow the dual delta calculation for GP but also consider the additional isotope effect from photosynthesis (see below).

For their calculations, Prokopenko et al. 2011 used $\delta^{18}O_P$ value of -23.320 ‰ (vs. VSMOW). Luz and Barkan 2011, GRL however showed that a small difference exists between the water values and the average phytoplankton for which they provide $\delta^{18}O_P$ −20.014 ‰. The difference between these two values, 3.306 ‰ for $\delta^{18}O_P$ and 26 per meg for $^{17}\Delta_{bio}$, therefore reflects the associated fractionation between the substrate water and the photosynthetic $O_2$. To obtain representative $\delta^{17}O_{P-FR}$ and $\delta^{18}O_{P-FR}$ values for the Feitsui reservoir, we assume the 26 per meg difference to our $^{17}\Delta$ of water and the 3.306 ‰ difference to the $\delta^{18}O$ of the Feitsui reservoir water and retrospectively calculate the $\delta^{17}O_{P-FR}$. The newly obtained $\delta^{17}O_{P-FR}$ and $\delta^{18}O_{P-FR}$ vary between -13.371 ‰ and -12.613 ‰, and -26.337 ‰ and -24.874 ‰, respectively throughout the sampling period, with the average $\delta^{17}O_{P-FR}$ and $\delta^{18}O_{P-FR}$ being -13.183 ‰ and -25.975 ‰.

***P9:***

*L09 – By DEC14 thermal stratification nearly disappeared" but mld became deeper? How is mld defined? It follows the color code of Chl a in Fig. 2 but this might be an artifact of the figure. How vertical mixing is high but a very strong marked mld? What happened from SEP14 to OCT14 that the stratification broke, which physical process dominated for this change in the water column? (horizontal or vertical transfer? Why? Wind speed increase? Rain?), increase in atm O2, but you cannot see that in δO2/Ar, I recommend using ΔO2/Ar instead.*

**Reply #17**

For further details on the mixed layer and mixing processes in the Feitsui reservoir please see response #3 and #1 above. We will include rainfall record and wind speed data in the revised manuscript and expand on the discussion of our results in context of these data (please see response #2 and #5). We thank for the suggestion and changed to expressing the $O_2$/Ar ratio as ΔO2/Ar, instead.

*L30 – is the nutrient availability also so high as in 100 m in spring 2015? Which physical process may dominate in the reservoir to achieve this?*

**Reply #18**

In the Feitsui reservoir, vertical mixing from changes in the mixed layer depth determines the nutrient availability for phytoplankton in the spring, while typhoon intensity is detrimental in the summer and autumn. The limiting nutrient for phytoplankton growth in the Feitsui reservoir is phosphate; the phosphtate data for spring 2015 are yet to be analysed. Irrespective of the concentration of nutrients at depth in spring 2015, it is very unlikely that the increased $^{17}\Delta$ at depth in APR15 and MAY15 represents local photosynthetic production. The $O_2$ concentration at 90 m was typically very minimal (also seen from DO and $\delta O_2/Ar$) and on few occasions only could be measured and should be interpreted with caution. Due to low $O_2$ amount at this depth, it is likely that only small contribution of water with different $^{17}\Delta$, could significantly affect the $^{17}\Delta$ signal at 90 m. Please see reply #6 for a likely process.

*L32 – How common are seiches in Feitsui Reservoir? This seems to be an important process that dominates the distribution of nutrients and gases vertically in the water column of the reservoir. More information is needed on this regard and the authors should discuss more this process in the context of their findings.*
*Which other external forcing processes the authors meant there?*

**Reply #19**
Please see above reply #6 where we also addressed seiches. Seiches were never studies in the reservoir so we are unable to give more details on this. As mentioned in the above reply #17, changes in the mixed layer drive vertical mixing in the Feitsui reservoir, which determine the vertical distribution of nutrients and gases in the spring, while typhoon occurrences play a key role in summer and autumn. In the revised manuscript we will include the precipitation record and wind speed data (please see replies #4 and #13) throughout the period of our study and will expand on the discussion of these processes with regards to our findings on dissolved oxygen and water variations.

*P10:*
*L1 – were precipitation events recorded during the sampling periods (not only the typhoon events)?*

**Reply:** Yes, please see above replies #2 and #5 for further details.

*L2-5 – here the authors suggest that vertical processes are relevant for the distribution of the 17Δ signal in the reservoir, it*

**Reply #20**
To quantify the effect of vertical mixing due to changes in thermal stratification we provide results from the column inventory approach. Please see reply #1 for further details.

*L11-12 – most of the production seems to take place below the mld, L12 – again this is arguable because of their definition of mld*

**Reply #21**
Please see above response #3 for the definition and further details on the mixed layer. Please see reply #1 with results from the column inventory approach, which enable us

to estimate the contribution of production from below the mixed layer.

*P11:*
*L6 – how was that lifetime of O2 estimated? Was based on mld and gas transfer coefficient?*

**Reply #22**
We thank the reviewer for pointing this out. Indeed, the term lifetime is here confusing. We refer to the mixed layer mixing time, obtained from the mixed layer depth and the gas transfer coefficient. Previously we provided a rapid estimate only, we have now corrected this to span the whole sampling period and clarified it in the revised manuscript. Using the mixed layer depth and the gas transfer coefficient we have determined the mixing time throughout the period of study to vary between minimum 6 days and maximum 54 days (estimated for JUL15 and JAN15, respectively) with the average residence time being about 24 ± 18 days.

*L7 – I disagree that the vertical mixing and advection are negligible for 17Δ and ultimately GP determinations in the reservoir. I think the authors underestimate here the vertical transfer of biological O2 (vertical displacement of primary producers within the reservoir) and their definition of mixed layer depth is by no means helpful for their budget model. The fact that there is high 17Δ in subsurface and deep waters defines the reservoir as a full column activity water system with marked seasonality and strong vertical influence, possibly due to wind and rain. It is hard to apply there the GP concept from δ17O and δ18O considering the shallow mixed layer depth. This is irrelevant to calculate here, I would be more focused on explaining the physical driving forces to the vertical distribution of 17Δ, and how this changes in short periods of time the metabolic balance of the reservoir.*

**Reply #23**
Please see reply #1 for further details on vertical mixing and results from the column inventory approach and reply #3 for further details on mixed layer. To properly associate the effect of wind and rain on changes in $^{17}\Delta$ and other findings we will include the record of precipitation and wind speed and discuss our data in context of these parameter. We will also expand the discussion on other physical processes (such as changes in thermal stratification) and their effect on the vertical distribution of $^{17}\Delta$ (please see replies #2, #5 and #6).

*L8 – but most part of the sampling period the bottom limit of the euphotic zone lies below the mixed layer, so again the authors should revisit their definition of mld for GP calculations*

**Reply:** Please see replies #1 and #3.

*L24-25 – more information is needed on the preparation of the air-equilibrated aliquots*

**Reply:** Please see reply #10.

*P12:*
*L09-10 – I agree, but then why the authors wrote lines 10-12 in P. 11? This is*

*contradictory to what is stated here. To actually compare productivity values based on 14C and from oxygen isotopes, the sampling scheme had to be designed for this purpose with duplicate analysis and samples for 14C also at depth.*

**Reply #24**
We thank for highlighting this issue and have now removed the sentence in lines 10-12 on P.11.

*Minor comments:*
*Throughout the manuscript, leave a space between the quantity and the unit in % and ‰, and also for °C* – changed
*P1, L15 – add "water" reservoirs* – added
*P2, L11 – change "confining it to a small volume" to "confining them into a small volume"* – changed
*P2, L16 – replace the symbol "&" by the word "and" here and all the citations throughout the manuscript where it is used* – we have replaced all "&" symbols by "and" throughout the manuscript
*Modify to "introduced the triple oxygen-isotopes technique, ..."* – modified
*P2, L17 – change to "The $^{17}O$ excess is defined as:"* – changed
*P2, L19 (eq. 1) – here and elsewhere all variables must be italicized, this is particularly the case of all delta symbols in: $^{17}\Delta$, $\delta^{17}O$ and $\delta^{18}O$, as well as K, Co introduced in eqs. 2 and 4, and throughout the manuscript.* – we have italicized all mentioned symbols throughout the manuscript
*P3, L3 – change "large" to "largely"* – changed
*P4:*
*L1 – add comma between "quality" and "the watershed" L1 – add "the" before Feitsui Reservoir* – added
*L2 – pluralize "area"* – done
*L2 – add "are" between "active" and "prohibited"* – added
*L3 – add "the" before Feitsui Reservoir* – added
*L4 – since when the meteorological station near Feitsui Reservoir has been active?*
The meteorological station at Feitsui Reservoir has been active since January 1988. We have included this in the amended manuscript.
*L5 – change from "processing" to "preparation"* – changed
*L9 – change to "using a Sea-Bird CTD...", what was the vertical resolution of the CTD measurements?*
We would like to correct this to "using Ocean Seven 316 CTD (IDRONAUT, Italy)", which is the sensor used during sampling in the Feitsui Reservoir (a Sea-Bird CTD was used during other cruises). The vertical resolution varied between 10 and 50 cm; depending on the weather conditions during the sampling.
*L14 – leave a space between the number and the units (15 μL)* – done
*L19 – "...for removal of water vapor at liquid nitrogen temperature. The extracted gases..."* – changed
*L21 – The GC is to separate $N_2$ and $CO_2$ from $O_2$ and Ar, please delete "contaminants". Please correct and complete the sentence by adding that only $O_2$ and Ar remain the main components in the gas mixture.*
We have corrected the sentence, it now reads "Extracted gases were then either stored in sealed glass tubes or directly connected to a GC system (Thermo Scientific TRACE Gas Chromatograph) for complete removal of $N_2$ after which only $O_2$ and Ar remained the main components in the gas mixture".

*L22 – "During the separation ... " –* changed

*L25 – I suggest here to add a new section (2.3) that corresponds to the "Stable isotope analysis in water". Also an opening sentence to explain why this was done is needed, for example: "To identify the source of water in the reservoir, the δD and δ¹⁸O in the H₂O molecule of reservoir water was analyzed. For this, water samples were collected in 15 ml ...."*

We have made the modifications as suggested; section "2.3 Stable isotope analysis in water" was added and the opening sentence now reads " To identify the source of water in the reservoir, the δD and $\delta^{18}$O in the H$_2$O molecule of reservoir water was analyzed. For this, water samples were collected in 15 ml centrifuged vials…"

*L30- change uL to mL and to "an aliquot of 5 mL of water sample was converted to O₂ by injecting it to a CoF₃ reaction tube...". Leave also a space between 370 and °C.* – corrected

**P5:**

*L3 – Do you mean that a set of duplicates of standard water samples were measured every 80 water samples analysis?*

Here, we meant 80 changeover valve changes, i.e., 80 cycles. We have now clarified this in the manuscript.

*L8 – change to "O₂ from the purified oxygen-argon mixture (as explained in section 2.2)..." –* changed

*L9 – change to "12 cycles each. Thus, the reported ..." –* changed, it now reads "Each sample was run for 3 acquisitions, 12 cycles each, thus the reported δ values present the average of 36 cycles"

*L18 – this precision is for δO₂/Ar in repetitions of atmospheric air measurements? –*

Yes, routine measurements of atmospheric air samples over the period of about 6 months (during which all reservoir samples were analyzed). We have now included this data to the supplement.

*L23 – the correction is not to achieve high precision, it is simply a correction of the measurement due to interferences. Delete this sentence. –* deleted

*L26 – remove the second "of" (..."and for obtaining more precise results...") –* corrected

*L29 – How many samples represent one set or trip to the reservoir? Are all the black dots plotted in a single vertical profile in Fig. 5 representing one set of samples? They were not always the same number isn't?*

One set – one trip to the reservoir typically represents 9 samples (1, 5, 10, 15, 20, 30, 50, 70 and 90 m depth), we have now clarified this in sec. 2.2. Yes, all the black dots in one vertical profile represent one set of samples. Often, samples from 70 and 90 m depth could not be measured due to very low concentration of oxygen (see P9 L15), few samples were lost in the field (bubbles were seen during filling or the flask broke) or during sample preparation.

**P6:**

*L2-8 – what is written in this paragraph is only true for a system at steady state. This should be stated.*

We thank the reviewer for pointing out. We agree with the reviewer that the statement is valid at steady state only and amended this in the manuscript.

*L16 – Co as expressed by the authors is not simply the O₂ solubility, but the O₂ concentration at saturation, or at equilibrium with the atmosphere, using the solubility coefficients from Benson and Krause, 1984, and the standard term to express this is [O₂]sat.*

We thank the reviewer for pointing this out and have modified it in the manuscript.

*L17 – did the daily wind speed measurements were collected from the meteorological station? Were they corrected to represent wind speed 10 m above sea level? Why averaged over 1 week? What is the residence time of the gas in the mixed layer depth of the reservoir as calculated from the gas transfer coefficient and the mixed layer depth?*

The wind speed data were collected from the Feitsui meteorological station that provides direct wind measurements 10 m above the water level. Because the gas concentrations in the mixed layer depend on the recent history of wind speeds, the appropriate period over which to average the wind speed corresponds to the $O_2$ residence time in the mixed layer. Previously we only provided a rough estimate, we have now corrected it and averaged the K over the residence time of $O_2$ in the mixed layer estimated preceding each sampling (please see above reply #22).

*L22-25 – this paragraph should be moved to another section maybe below section 2.2, stating specifically how the $^{14}C$ analysis was done.* We have moved this to a new section, which describes the $^{14}C$ analysis.

*L29 – change to "... Ar supersaturaion in water ..."* – changed

**P7:**

*L2, Eq. 3 –The term on the left hand side of Eq. 3 is misleading and doesn't represent what is really expressing. The biological $O_2$ saturation should be expressed as $\Delta O_2/Ar$ as in many past works that use this method (e.g. Cassar et al., 2011, Castro-Morales et al., 2013). Please avoid introducing new ways for terms and variables. The new community using this method should make use of the same variables to express the terms to avoid confusion and to keep consistency.*

We have made the modifications as suggested and expressed the $O_2/Ar$ ratio as $\Delta O_2/Ar$.

*L6-7 - here it should be stated that the authors corrected $\delta O_2/Ar$ for the residual gas in water sample after equilibration in order to obtain their $([O_2]/[O_2]_{eq})_{bio}$ (that should be $\Delta O_2/Ar$) in eq. 3.*

We have included this in the revised manuscript; please also see reply #13 above.

*L21 – I wouldn't call it permanent but seasonal stratification* – we have amended the terminology

*L22 – the temperatures above 30 $^oC$ are only in the top 10 m.* – clarified

*L25 – here it should be defined which criterion the authors used to define mixed layer depth.* – we will include the definition (please see reply #3)

*Is only the change in atmospheric temperature what makes the temperature of the water reservoir to change? Is there no evidence of vertical or horizontal water transport? As mentioned later in the manuscript, other lake processes as the presence of seiches (P9, L32) or other external influences such as wind or water input from precipitation can also alter the temperature of the reservoir. Discuss this here in the context of this factors possibly contributing to the change in the water temperature. Of particular interest is what happens at depth in the reservoir, away of the direct atmospheric influence.* – We will add the discussion here as suggested by the reviewer, please see also our responses #1, #2 and #5.

*L26 – which processes occur within the reservoir to shallow the mixed layer depth in summer? Only warming by atmospheric influence at the surface?*

Although during the summer months the reservoir is strongly stratified (in particular as of July) as a result of continued heating of the surface water, rainfall and windstorms reach to a depth of about 10 to 15 m (can be seen from dissolved $O_2$ saturation for example) encompassing the mixed layer and the upper thermocline.

These processes therefore play prevailing role and influence the conditions in the mixed layer.

*L27 – in all other cases where mixed layer was present was also defined, only shallower –* we have rephrased this

*L28 – where are these sediments from? From the bottom or from lateral transport within the reservoir? Which other lake processes? –* We have not evaluated in detail the sediment suspended in the water and will remove this postulation from the sentence. Please see reply #6 for suspended materials in water column.

*P8:*

*L13-16 – during late spring in 2014, $O_2$ supersaturation also at the bottom of the reservoir is seen, what is the origin of this? It should be then a very strong vertical mixing in the reservoir at this period of time, this is the indication of a vertical transfer also of potentially biological $O_2$. Why is so fast changing this to a very shallow $O_2$ supersaturation by end of April?*

We did not report $O_2$ supersaturation at bottom, nor showed in the figures. Please see Reply #6 for occasional occurrence of high $O_2$ content at depth.

*L22 – complete the sentence as: "showed more depleted values during autumn at the top 60 m..." –* done

*L23 – what do the authors mean with "selected waters"? –* please see reply #15

*P9:*

*L5 – wouldn't be better JUL14 than JUN14? –* we do not have samples for JUL14, only CTD data is available, we have made description more clear now.

*L7 – Also in JUL14 –* please see above comment

*L8-9 – the thermal stratification nearly disappeared, but the mixed layer depth just became deeper, how it is defined mld? –* we have rephrased this description

*L22 –how much is the annual mean?* The mean surface $^{17}\Delta$ was 59 ± 13 per meg

*P10:*

*P10, L17-18 – remove one dot at the end of the sentence (it has two) and the dot at the end of line 18 should only have the comma. –* removed

*L6-7 – change to "...we briefly discuss our results in the context of typhoon events." –* changed

*L18 – higher wind speeds can also explain higher GP rates at depth of the reservoir? –* We have only measured GP rates in the mixed layer (and the euphotic zone using $^{14}C$ method).

*References*

*- The reference of Barkan and Luz, 2005 is missing*

**Reply:** We have included the reference in the manuscript now.

*Figures:*

*Fig. 2, is one monthly band of data representing only a once in a month sampling? So this is not really an entire month of data but only few days (maybe only one day) in which sampling a set of samples took place? The interpolated figures as shown in Fig. 2, 3 and 5 are then very misleading, since the vertical data cannot be put sequentially one after the other and do a horizontal interpolation with them. There is a gap of about 29 days between them, and as it looks now in the figures, "fast" changes occur between one month and the other. I will be careful in the way the data is presented in this figures. I would rather do simply vertical profiles or not put together the bands. It is unclear to see how much of the information on the figure is the result of interpolation artifacts.*

*Also, the mixed layer depth and limit of euphotic zone should be drawn also from May 2014.*

**Reply:** In Fig. 2 one band represents typically one week, exactly CTD casts were carried out on weekly basis during the summer months and every two weeks during the winter. This regular sampling is part of a long-term study run by the Environmental Ecosystem Laboratory group and therefore variability in temperature, chlorophyll and dissolved $O_2$ is well constrained for the past years. Based on comparisons with data from previous years (Itoh et al. 2015) it is highly unlikely that plots in Fig. 2 are a result of interpolation artifacts. Sampling for isotope analyses (dissolved oxygen and water) was usually carried out on monthly basis (Fig. 3 and 5), with the exception of two occasions when more than a month is between the sampling points (between JUN14 and AUG14, and FEB15 and APR15). We thank the reviewer for the suggestion and have checked the interpretation based on the obtained CTD profiles, which are available at higher sampling rate. We believe that the smoothed/interpolated figures remain representative and consider them a better way to visualize the data in context of the temporal variability. However, to make the data visualization approachable to all readers we have decided to include the simple vertical plots along with the raw data in the supplementary material and will refer to it in the figure description.

*Fig. 3, what happened with the data from May to August 2014?*
*The lower $\delta^{18}O$ and $\delta D$ from the surface to about 60 m from October to December 2014 is related to the first typhoon according to the authors, However, at the time of the second typhoon there is also a different $d^{18}O$ signal at the surface (top 20 m in May-July 2015), the authors must explain these differences and linkages to $\delta^{18}O$ and $\delta D$ from dissolved oxygen as shown in Fig. 5a for the second typhoon.*

**Reply:** The sampling for water isotopes was initiated later than the dissolved oxygen routine thus no data is available prior to September 2014. To keep all plots comparable we did not amend the x-axis and rather left the first few months blank. The effect of typhoon events on the isotopic composition of dissolved oxygen and water in the reservoir is yet to be fully understood and any conclusion may be misleading due to insufficient data. Although this is beyond the scope of our present study, we are currently working on a more rigorous sampling effort which samples are yet to be analyzed and results will be presented in the future. However, as already mentioned above we have expanded on the linkages between the isotopic composition of dissolved oxygen, water and physical processes in the revised manuscript and will also will also discuss the data in the context of meteorological data obtained form the Feitsui reservoir.

*Fig. 5,*
*5b, the low $^{17}\Delta$ in the water column from 20 m down during May-June 2014 is coincident with very high $O_2$ saturation, which indicates a strong vertical mixing from surface air saturated water down. This is also evidenced in the $\delta O_2/Ar$ signal. However, the authors claim that the high $^{17}\Delta$ seen at the depth during July-October 2014 also originates from vertical transfer from the surface, however, there is lower $^{17}\Delta$ signal in the top <10 m. could it be local $O_2$ photosyntetically produced? or horizontal transfer? Which process actually causes breaking down the high $^{17}\Delta$ in the entire column between 40 and 60 m from July-November 2014?*

**Reply:** We thank the reviewer for highlighting this point. The low $^{17}\Delta$ in the water column below 20 m during JUN14 is likely indicative of strong vertical mixing of air-saturated water down the water column (also supported by dissolved $O_2$ saturation and $\delta O_2/Ar$), as a result of heavy rainfall. After an analysis of newly acquired data from the Feitsui Administration Bureau we observe about ~700 mm of accumulated precipitations during JUN14, comparing to ~340 and ~200 mm measured in JUL14 and AUG14, respectively. However, it is important to stress that because of this being the sole occasion when such trend is observed further observations are necessary before concrete conclusions may be drawn. As already mentioned above, we will include all precipitation data for the sampling period in the amended manuscript.

Following from August to October 2014 accumulation of $^{17}\Delta$ is seen below the thermocline, however with distinct $\delta O_2/Ar$ signal during the first half (AUG14 and early SEP14) and the second half (late SEP14 and OCT14) of this period. While earlier this is indicative of high local primary production possibly a phytoplankton bloom, later we see a decrease in $O_2$ concentrations with high $^{17}\Delta$ signal suggesting $O_2$ consumption as well as the absence of vertical mixing and any contribution from atmospheric air. Between 40 – 60 m depth, from August to October 2014, the low $^{17}\Delta$ signal counter correlates with dissolved $O_2$ saturation, with lower $^{17}\Delta$ and higher DO observed in AUG14 and SEP14 and increasing $^{17}\Delta$ and decreasing DO saturation observed in OCT14. Additionally, the signal also follows the thermal structure of the reservoir; from July to about November 2014 we observe well mixed epilimnion in the upper ~10 to 20 m and an extensive metalimnion to about 50 – 60 m, with strong thermal gradient before reaching hypolimnion below. It is likely that the low $^{17}\Delta$ origins from atmospheric air entrainment, especially during the early summer (see also DO for June and July) which is confined to 40 – 60 m due to the strong thermal gradient and is not altered by photosynthesis due to the lack of primary producers in this region. The breaking down of this signal in October and November is then controlled by decreasing air temperatures and weakening thermal stratification towards the winter overturn in DEC14. In the revised manuscript we have expanded the discussion on the physical processes and related changes in $^{17}\Delta$.

*First signal of high $^{17}\Delta$ at the bottom (80-100 m) in March-May 2015 it seems is a different water mass, this is also seen in the $\delta^{18}O$ and $\delta O_2/Ar$, what is its origin? It looks lateral transport.*

**Reply:** Please see above reply #6 on the surface water intrusion and reply #18 where we also addressed the high $^{17}\Delta$ near the bottom of the reservoir. Please note that the 90 m samples are often limited due to very low $O_2$ concentration in the samples, and therefore it is very difficult to postulate on the origin of the $^{17}\Delta$ signal at 90 m.

*5c, why is this third depth point at around 20 m in August 2014 so high in O2/Ar? Most of $\delta O_2/Ar$ is below zero. It is hard to see the biological and atmospheric contribution in this ratio. A better way to express this is as $\Delta O_2/Ar$ (biological $O_2$ saturation) in % (this is their $([O_2]/[O_2]_{eq})_{bio}$). I recommend the authors to plot instead $\Delta O_2/Ar$ in Fig. 5 panel c.*

**Reply:** We thank for the suggestion and as already mentioned in the above responses we have now used the $\Delta O_2/Ar$ instead in the revised manuscript. We closely inspected

the data point mentioned (20 m, AUG14) and could not find any obvious issues here. The intensity on m/z 40 was much lower than usually, signifying lower Ar amount in the sample. Since this value is more than by a factor 20 higher than the second highest one measured, and vertical profiles of DO and $\delta^{18}O$ do not indicate any photosynthetic activity that could be proportional to these changes, it is likely that this data point is erroneous. One possible way to obtain low amount of Ar in sample is due to incomplete absorption of Ar during the GC separation. This is however highly unlikely to occur, firstly because the Ar-$O_2$ mixture is being trapped right front the start when the sample is sent to the chromatographic column, and secondly because we routinely measure atmospheric air samples that did not show any discrepancies in the $O_2$/Ar ratio. Based on this, we have decided to exclude the data point from the figure, however we have indicated this in the description of the figure and will also provide the value in the supplementary material.

*Fig. 6, are the $^{17}\Delta$ GP shown there is only the surface values?*

**Reply:** In Fig. 6 we show the integrated $^{17}\Delta$ GP for the mixed layer. We have now amended the figure description for clarification.

**Response to reviewers' comments on manuscript bg-2016-49 ("Variations in triple isotope composition of dissolved oxygen and primary production in a subtropical reservoir") by Jurikova H., T. Guha, O. Abe, F.-K. Shiah, C.-H. Wang and M.-C. Liang**

**Reply to comments by reviewer #2**

*The authors report measurements of $O_2/Ar$, $^{17}O/^{16}O$ and $^{18}O/^{16}O$ ratios of dissolved gases in a fresh water reservoir. They sampled the water column for more than a year. In addition to dissolved gases, they also measured $^{18}O/^{16}O$, D/H and $^{17}O$ excess of water. Using these measurements they estimated gross and net primary production and their ratios (GP, NP and NP/GP respectively). In their estimates they applied the method introduced by Luz and Barkan in 2000 (LB00). LB00 demonstrated the potential of the method for both marine and fresh water studies. Since then the method has been used a number of times in marine systems but not in fresh water ones, so the data set collected is valuable in that it adds information on triple oxygen isotope variations in a freshwater system. This information has potential to help understanding the metabolic balance in lakes and fresh water reservoirs. Yet, there are a number of issues that need to be addressed before the material in the manuscript is suitable for publication.*
*In order to meaningfully interpret the results in quantitative terms of GP and NP, the authors need to realize that the LB00 method is applicable to mixed layer which is at steady state with respect to fluxes of photosynthesis, respiration and gas exchange with negligible effects of vertical and horizontal advection.*

**Reply:** Please see reply #1 above where these concerns are addressed.

*While these conditions may be assumed for a number of marine situations, the reservoir in this manuscript may be more dynamic. If that's the case, to obtain meaningful quantitative estimates of GP and NP, the authors will need to include at least some of such dynamics in their calculations and apply a non-steady state model. While this may be a tall order, at the least, such approach should considered and discussed and the present estimates should be qualified and treated in a qualitative way. The data base of the study should be made available for future studies (see below) when a non-steady state model becomes available.*
*The authors are aware that a portion of the reservoir's photosynthesis takes place in the photic zone beneath the mixed layer. They have to give an estimate of how much is missing in their estimates for the mixed layer.*

**Reply:** Please see reply #1 above and the whole column inventory approach for an estimate of the degree of the robustness of the mixed layer model.

*As well, in order to apply the LB00 method, it is necessary to know the $^{17}O$ excess of photosynthetic oxygen. While the latter depends on the $^{17}O$ excess of water, the two are not identical and the difference may be significant (see Luz and Barkan, 2011, GRL).*

**Reply:** We thank the reviewer for pointing this out; please see our reply #16 above. The difference between the respective GP rates obtained with different $^{17}\Delta_{bio}$ is illustrated in the figure in our reply #1 above.

*Even if the difference between $^{17}O$ excess of photosynthetic and water oxygens is known, I expect the value for water in the reservoir to be variable and to be dependent on fluctuations in the isotopic composition of meteoric water and evaporation from the reservoir. So more measurements of $^{17}O$ excess of water are needed. The authors give one value for $^{17}O$ excess of water (246 per meg with respect to air). What are its $d^{17}O$ and $d^{18}O$ values?*

**Reply:** Please see our replies #15 and #16 above. Although we do not expect high variations in the $^{17}\Delta$ of the water, due to small variations in $\delta^{18}O$ and $\delta D$, and long residence time (~150 days) of the reservoir water, we are presently analysing more water samples archived that will enable us to fully constrain any potential seasonal variations in $^{17}\Delta$ of the water. The measured values for the water sample mentioned were -15.031‰ and -29.275‰ vs. to air for $\delta^{17}O$ and $\delta^{18}O$, respectively.

*Importantly, all raw data for $d^{17}O$ and $d^{18}O$ of dissolved and water oxygen should be given in tables suitable for web appendix if the paper is published.*

**Reply:** We have included supplementary material to the revised manuscript with all raw sample data used in the study.

---

## Author Response (AR2)

Dear Dr. Middelburg,

We are pleased to hear that our manuscript has been accepted for publication in Biogeosciences. On behalf of all co-authors, thank you very much for handling our manuscript and identifying parts that needed further corrections; these have now been revisited and corrected as follows: we have made all technical corrections to the text as suggested, added an explanation for PB-I to the reader, revised the reference list to ensure all references are included and renumbered the figures so that they now follow a sequential order according to citations in text.

Please find our marked-up manuscript below.

Yours sincerely,

Hana Jurikova
GEOMAR Helmholtz Centre for Ocean Research Kiel

[revised manuscript text omitted]

---

## Author Response (AR3)

Dear Dr. Middelburg, BG Publication Office Ms. Santana,

I first thank you for handling and editing our manuscript and apologize for taking so long to do the proof reading / correction.

I am extremely sorry that during the final stage of proofreading the manuscript, we found some numbers in our calculation sheet were misplaced, that results in quite a few corrections needed for the derived GP and NP values and figures, though major conclusion remains unchanged. Additionally, some $^{17}\Delta$ values for dissolved $O_2$, and the $^{17}\Delta$ of water were amended, however, these remain the same (within error) as the values used earlier. I consider this as a major correction that requires further approval from the handling editor. Below, please find the marked-up document, highlighting the new corrections made to manuscript submitted for typesetting. The revised document also includes changes we made following the remarks from the language copy-editor and the typesetter.

I sincerely apologize for any inconvenience. Please let me know your decision and advise us what we should do next. If this version of correction is accepted, we will incorporate the changes to the journal edited version.

Sincerely yours,

Mao-Chang Liang

[revised manuscript text omitted]